# CRISPR-enabled point-of-care genotyping for *APOL1* genetic risk assessment

Robert Greensmith [1,2], Isadora T Lape[3], Cristian V Riella[4,5], Alexander J Schubert [1,2,6],
Jakob J Metzger [1], Anand S Dighe[5,7], Xiao Tan[8,9,10,11], Bernhard Hemmer[12,13], Josefine Rau [1],
Sarah Wendlinger[1,2], Nora Diederich[1,2], Anja Schütz [14], Leonardo V Riella [3,15,16 ✉] &
Michael M Kaminski [1,2,6 ✉]

## Abstract

Detecting genetic variants enables risk factor identification, disease screening, and initiation of preventative therapeutics. However, current methods, relying on hybridization or sequencing, are unsuitable for point-of-care settings. In contrast, CRISPR-based-diagnostics offer high sensitivity and specificity for point-of-care applications. While these methods have predominantly been used for pathogen sensing, their utilization for genotyping is limited. Here, we report a multiplexed CRISPR-based genotyping assay using LwaCas13a, PsmCas13b, and LbaCas12a, enabling the simultaneous detection of six genotypes. We applied this assay to identify genetic variants in the *APOL1* gene prevalent among African Americans, which are associated with an 8–30-fold increase in the risk of developing kidney disease. Machine learning facilitated robust analysis across a multicenter clinical cohort of more than 100 patients, accurately identifying their genotypes. In addition, we optimized the readout using a multi-analyte lateral-flow assay demonstrating the ability for simplified genotype determination of clinical samples. Our CRISPR-based genotyping assay enables cost-effective point-of-care genetic variant detection due to its simplicity, versatility, and fast readout.

**Keywords** Biomarkers; Diagnostics; CRISPR-based-genotyping; APOL1; Kidney Disease
**Subject Categories** Biotechnology & Synthetic Biology; Genetics, Gene Therapy & Genetic Disease; Methods & Resources

## Introduction

CRISPR diagnostics enable the sensitive and specific detection of DNA and RNA biomarkers at the point-of-care (Gootenberg et al, 2017; Abudayyeh et al, 2017; East-Seletsky et al, 2016; Kaminski et al, 2021). Single nucleotide specificity of certain Cas enzymes, such as Cas12 or Cas13 make them promising tools for rapid genotyping in point-of-care settings (Kumar et al, 2022). However, while these methods have predominantly focused on pathogen detection (Mustafa and Makhawi, 2021; Broughton et al, 2020; Dai et al, 2019; Kaminski et al, 2020), their potential for single nucleotide variant sensing has yet to be fully harnessed. Current CRISPR-based genotyping assays mainly rely on two guide RNAs to distinguish targets that differ by a single-base pair, yet they face challenges in robustly detecting heterozygous alleles. Further, these methods have not yet been optimized for multiplexing, which would allow for the simultaneous detection of multiple genetic variants in a single reaction. Notably, the potential of CRISPR-based genotyping remains unexplored in larger clinical cohorts, impeding the validation and direct comparison of these methods with current clinical practices. Therefore, our study aims to address these gaps by investigating the performance of a multiplexed CRISPR-based genotyping assay in a large clinical cohort and adapting it toward a point-of-care test.

The disproportionate burden of non-diabetic kidney diseases among individuals of sub-Saharan African ancestry is evident, with a four-fold higher risk of developing end-stage kidney disease than those of Asian and European ancestry. Initially, socioeconomic factors were proposed as the primary drivers of this disparity (Siemens et al, 2018). However, the identification of high-risk variants (G1 and G2) of Apolipoprotein L1 (*APOL1*) has challenged this notion, revealing a genetic susceptibility associated with the *APOL1* gene as the key determinant (Genovese et al, 2010).

[1]Berlin Institute for Medical Systems Biology, Max Delbrück Center for Molecular Medicine in the Helmholtz Association, Berlin, Germany. [2]Department of Nephrology and Medical Intensive Care, Charité Universitätsmedizin Berlin, Berlin, Germany. [3]Center for Transplantation Sciences, Massachusetts General Hospital/Harvard Medical School, Boston, MA, USA. [4]Nephrology Division, Department of Medicine, Beth Israel Deaconess Medical Center, Boston, MA 02215, USA. [5]Harvard Medical School, Boston, MA 02115, USA. [6]Berlin Institute of Health, Berlin, Germany. [7]Department of Pathology, Massachusetts General Hospital, Boston, MA, USA. [8]Wyss Institute for Biologically Inspired Engineering, Harvard University, Boston, MA, USA. [9]Division of Gastroenterology, Massachusetts General Hospital, Boston, MA, USA. [10]Institute for Medical Engineering and Science and Department of Biological Engineering, Massachusetts Institute of Technology, Cambridge, MA, USA. [11]Infectious Disease and Microbiome Program, Broad Institute of MIT and Harvard, Cambridge, MA, USA. [12]Department of Neurology, Klinikum rechts der Isar, Technical University of Munich, Munich, Germany. [13]Munich Cluster for Systems Neurology (SyNergy), Munich, Germany. [14]Protein Production & Characterization, Max Delbrück Center for Molecular Medicine in the Helmholtz Association, 13125 Berlin, Germany. [15]Department of Surgery, Massachusetts General Hospital/Harvard Medical School, Boston, MA, USA. [16]Division of Nephrology, Massachusetts General Hospital/ Harvard Medical School, Boston, MA, USA. ✉E-mail: lriella@mgh.harvard.edu; michael.kaminski@mdc-berlin.de

Carrying two risk variants (G1G1, G1G2, or G2G2—high-risk genotypes) substantially increases the risk of kidney diseases, including hypertension-associated end-stage kidney disease (H-ESKD), focal segmental glomerulosclerosis (FSGS), and HIV-associated nephropathy (HIVAN) (Genovese et al, 2010; Friedman and Pollak, 2020a; Kopp et al, 2011; Kasembeli et al, 2015). Furthermore, individuals receiving a high-risk genotype donor kidney exhibit worse kidney transplant outcomes and an increased likelihood of transplant failure (Julian et al, 2017).

Of significant relevance is the high *APOL1* risk allele frequency, estimated to be around 35% in the African American population in the United States, with ~13% of individuals within this group carrying two risk alleles (Friedman and Pollak, 2020b). Presently, *APOL1* genotyping involves PCR followed by Sanger sequencing, with commercially available tests taking up to three weeks to deliver results (Appendix Table S3). Furthermore, timely determination of *APOL1* genotype in donor kidneys would allow better risk stratification post-transplantation, complementing donor ethnicity from the current risk calculator (Julian et al, 2017). Currently, ethnicity is employed as a rudimentary indicator of *APOL1*-mediated kidney disease risk in clinical practice. However, this approach proves inadequate as the majority of individuals with recent African ancestry do not carry the high-risk genotype.

In this study, we developed a CRISPR-based, single reaction assay for accurate and fast determination of a patient's *APOL1* genotype, encompassing all six possible genotypes (G0/G0 or wild-type, G0/G1, G1/G1, G0/G2, G1/G2, G2/G2) (Fig. 1A). We leveraged the orthogonal cleavage properties of LwaCas13a, LbaCas12a and PsmCas13b to achieve multiplexed genotyping, and applied machine learning modeling for robust analysis that allowed for the test to be performed at two different centers with different personnel, machines, and reagent batches. We further validated this assay in a multicenter clinical cohort, and adapted it to a multi-analyte lateral-flow-based readout achieving diagnostic accuracy comparable to sequencing. While we demonstrate the use of CRISPR diagnostics for *APOL1* genotyping, the assay design, readout, and analysis can be broadly applied to other genetic risk factors and diseases.

Overall, we anticipate that rapid and accessible CRISPR-based genotyping will facilitate the identification of genetic risk carriers, increasing awareness of disease risk and enabling early genotype-guided therapeutic interventions, such as *APOL1* inhibitors (Egbuna et al, 2023). Moreover, its quick turnaround time and user-friendly nature make it suitable for multiplexed genotyping in emergency settings, during kidney transplant allograft stratification, in resource-limited settings and at the point-of-care.

## Results

We first defined the targets of our CRISPR-based *APOL1* (Data ref: NCBI Sequence Read Archive NM_003661.4, 2019) genotyping assay. G1 codes for two amino acid substitutions, S342G and I384M [dbSNP (Single Nucleotide Polymorphism Database) numbers rs73885319 and rs60910145] (Friedman and Pollak, 2020a). These SNPs are in almost perfect linkage disequilibrium, but ~1% of haplotypes with the S342G allele do not contain the I384M allele (Limou et al, 2015). However, the risk of disease in cases where the S342G allele is present, but not I384M, is believed to be the same as

when both alleles are present. Therefore, testing for S342G only is considered acceptable when assessing disease risk associated with the G1 variants (Friedman and Pollak, 2020b; Kopp et al, 2011; Thomson et al, 2014). We thus decided to sense the A > G SNP rs73885319 when detecting the G1 variant. The second variant associated with kidney disease and detected in our assay, G2 (rs71785313), is a six-base-pair in-frame (TTATAA) deletion that results in the loss of the amino acid residues 388N and 389Y (Friedman and Pollak, 2020a).

## Optimization of CRISPR-based detection of *APOL1* genetic variants

In order to identify suitable CRISPR RNAs (crRNAs) to genotype for the *APOL1* G1 and G2 variants, we performed a crRNA screen, systematically exploring the effect of the spacer sequence and its positioning. We first designed crRNAs for LwaCas13a and PsmCas13b and tested their ability to discriminate between RNA standards containing either target- or non-target alleles (Fig. 1). We designed nine LwaCas13a G1 sensing crRNAs such that the A > G SNP was placed at position 3 of the spacer sequence, with position 1 being the most proximal to the crRNA direct repeat sequence, and either one or two synthetic mismatches were introduced in proximity to the SNP to increase the crRNA's specificity (Fig. 1B). We found that the resulting signal intensities were greatly affected by the positioning and number of synthetic mismatches, and we identified a crRNA with two synthetic mismatches at positions one and four that showed the highest signal intensity ratios between target allele and non-target allele (crRNA 5, Fig. 1C). To genotype for the G2 variant, we next designed 26 different PsmCas13b crRNAs complementary to the wild-type (wt) sequence (TTATAA) (Fig. 1D). The position of adjacent spacer designs differed by one base, ranging from their 5' to 3' end. While the 6 base pair deletion (delTTATAA) as found in the G2 variant resulted in low non-target allele signal intensities for most crRNAs tested, we observed a large variance in the target allele signal intensities depending on the positioning of the spacer (Fig. 1E). Nine successive crRNAs (crRNAs 16–24) each resulted in a relative target allele signal intensity greater than 10, while the remaining 17 crRNAs resulted in a reduced target allele signal intensity. We thus selected the best performing crRNA 22 for further experiments. These findings suggest that synthetic mismatches can enhance crRNA specificity but reduce target allele activity when detecting point mutations, requiring optimization of both parameters. However, deletions can be adequately detected without additional mismatches, making the performance of crRNAs dependent solely on their target allele activity.

We next optimized the recombinase polymerase amplification (RPA) reaction to enable isothermal preamplification of genomic DNA (gDNA) to a concentration detectable by the Cas-crRNA complexes. We first tested primer combinations that would generate amplicons containing either the G1 SNP (Fig. 2A) or the G2 six-base pair deletion (Fig. 2B). Nine primer pairs were designed for each variant and tested for their ability to amplify synthetic DNA standards at either 1 pM or 10 fM, generating amplicons ranging between 103 and 197 base pairs (bps). We found G1 amplification to be more efficient than G2 amplification. While five of nine G1 primer pairs robustly amplified the 10 fM DNA target (Fig. 2A), only one G2 primer pair (F5R5) resulted in efficient

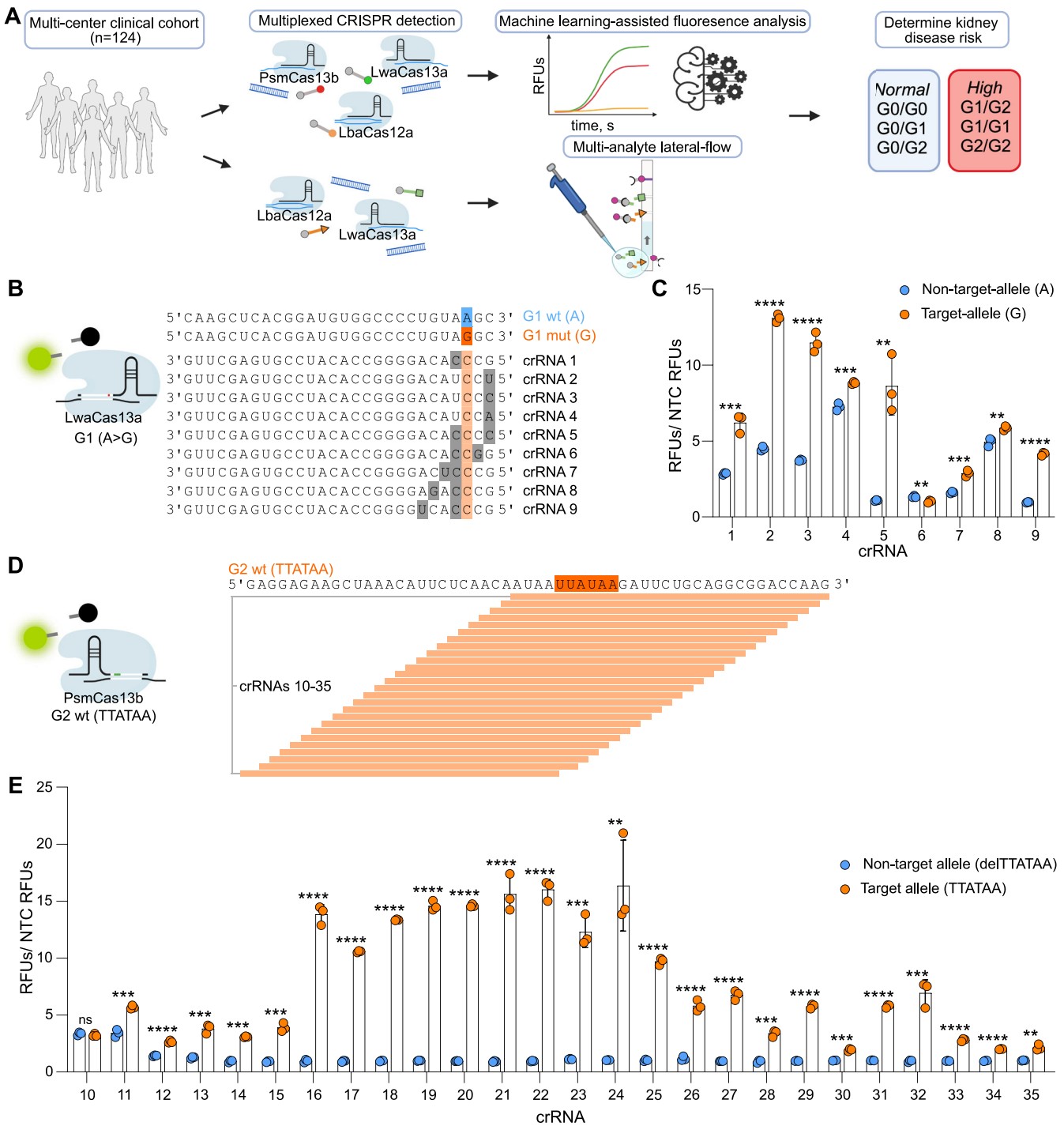

**Figure 1.  crRNA optimization for the rapid detection of *APOL1* risk variants.**

(A) Schematic of the assay described in this study. CRISPR detection of the *APOL1* risk variants is combined with either a fluorescence-based readout integrating machine learning for robust analysis or a multi-analyte lateral-flow-based readout enabling visual result interpretation. Differentiation of six *APOL1* genotypes enables the determination of kidney disease risk. (B) Alignment of target sequences and LwaCas13a G1 mutant (mut) sensing crRNAs. The G1 wild-type (wt) sequence is highlighted in blue and the G1 mut sequence is highlighted dark orange. Bases in the crRNA that are only complementary to the G1 mut target are highlighted in light orange. Synthetic mismatches complementary to neither G1 mut or G1 wt are highlighted gray. (C) Screen for G1 mut sensing LwaCas13a crRNAs using synthetic RNA mimicking G1 wt targets (blue) and G1 mut targets (orange). (D) Alignment of target sequence and PsmCas13b G2 wt sensing crRNAs. Dark orange highlights the position of the 6 bp G2 wt sequence. Light orange indicates complementarity between the crRNA and its target. (E) Screen for G2 wt sensing PsmCas13b crRNAs using synthetic RNA mimicking G2 mut targets (blue) and G2 wt targets (orange). (C, E) Synthetic target RNAs were detected at an overall concentration of 15 nM and each data point represents an individual reaction. For each reaction, RFUs are shown at 3 h divided by the mean RFU value of the NTC. Error bars: s.d. Statistical significance assessed with unpaired *t* test; $P > 0.05$= not significant (ns), *$P \le 0.05$, **$P \le 0.01$, ***$P \le 0.001$ and ****$P \le 0.0001$. For $P$ values, see Appendix Table S2. Source data are available online for this figure.

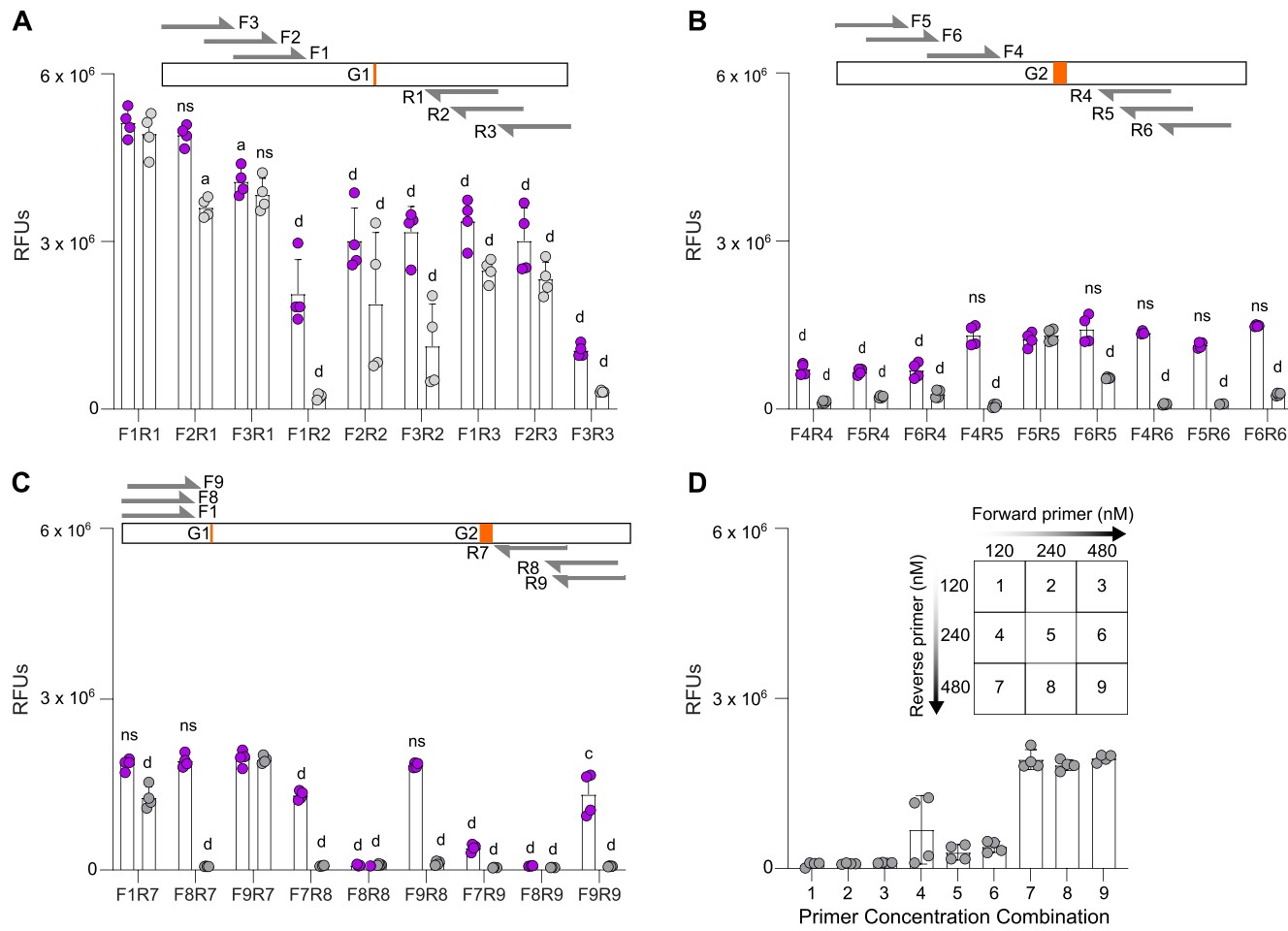

**Figure 2. Optimization of RPA design for isothermal amplification.**

(A–C) Schematics above each graph show different forward (F) and reverse (R) primers tested in relation to the target to be amplified. The position of G1 and G2 variants are shown in orange. RFUs shown at 3 h and each data point represents an individual reaction. Determining optimum RPA primer pair for amplification of DNA targets at an overall concentration of 100 fM (purple) and 2.5 fM (gray) for the G1 region (A), G2 region (B), G1 and G2 regions (C). (D) Primer optimization matrix assessing the indicated concentrations of forward and reverse primers. Primer pair used is F9R7 from (C). Each data point represents an individual reaction. (A–D) Error bars: s.d. (A–C) Statistical significance assessed with one-way ANOVA and Tukey's multiple comparison's test for 100 fM and 2.5 fM targets as compared to optimum primer pair; $P > 0.05 =$ not significant (ns), $P \leq 0.05$ (a), $P \leq 0.01$ (b), $P \leq 0.001$ (c), $P \leq 0.0001$ (d). Source data are available online for this figure.

amplification of both the 10 fM and 1 pM targets, albeit at lower signal intensities (Fig. 2B). To enable a multiplexed assay for simultaneous genotyping of both G1 and G2, we hypothesized that achieving equally efficient amplification of both regions would be advantageous for generating equimolar amounts of RNA for CRISPR detection. Therefore, we next tested if a single amplicon covering both variants could be generated by RPA (Fig. 2C). Elongating the amplicon did not reduce RPA efficiency, and two primer pairs were identified that robustly amplified both the G1 and G2 regions (F1R7 and F9R7) at 1 pM or 10 fM (Fig. 2C). F9R7 generated amplicons of 234 bps and showed the highest signal intensity. We further tested different forward and reverse primer concentrations (Fig. 2D) and found that a forward primer concentration of 120 nM together with a reverse primer concentration of 480 nM resulted in the most efficient amplification, which was selected for our assay.

To evaluate the capability to detect DNA targets in a multiplexed assay, we included the previously identified RPA primers and

crRNAs sensing the G1 mutant (LwaCas13a) and the G2 wt (PsmCas13b). As LwaCas13a and PsmCas13b detect RNA, the RPA forward primers contained a T7 promoter overhang for the transcription of DNA amplicons to RNA by T7 polymerase to enable subsequent CRISPR detection. We simultaneously measured the collateral cleavage activity of both LwaCas13a and PsmCas13b by using two RNA oligonucleotides (Gootenberg et al, 2018) containing a red and green fluorescent dye (Texas Red and 6-Carboxyfluorescein (6-FAM), respectively) that indicate orthogonal collateral cleavage preferences of these two Cas enzymes. We first tested the ability of the assay to detect and discriminate between synthetic DNA at 1 pM that mimicked three different *APOL1* genotypes: G2G2, G1G2, G1G1 (Fig. 3A,B). LwaCas13a and PsmCas13b both resulted in the highest signal intensity for the detection of the synthetic G1G1 (sG1G1) target, which contains the target allele for both enzymes. This was followed by a medium signal intensity for the heterozygous allele sG1G2, and the lowest signal intensity, similar to the non-template control (NTC), for the

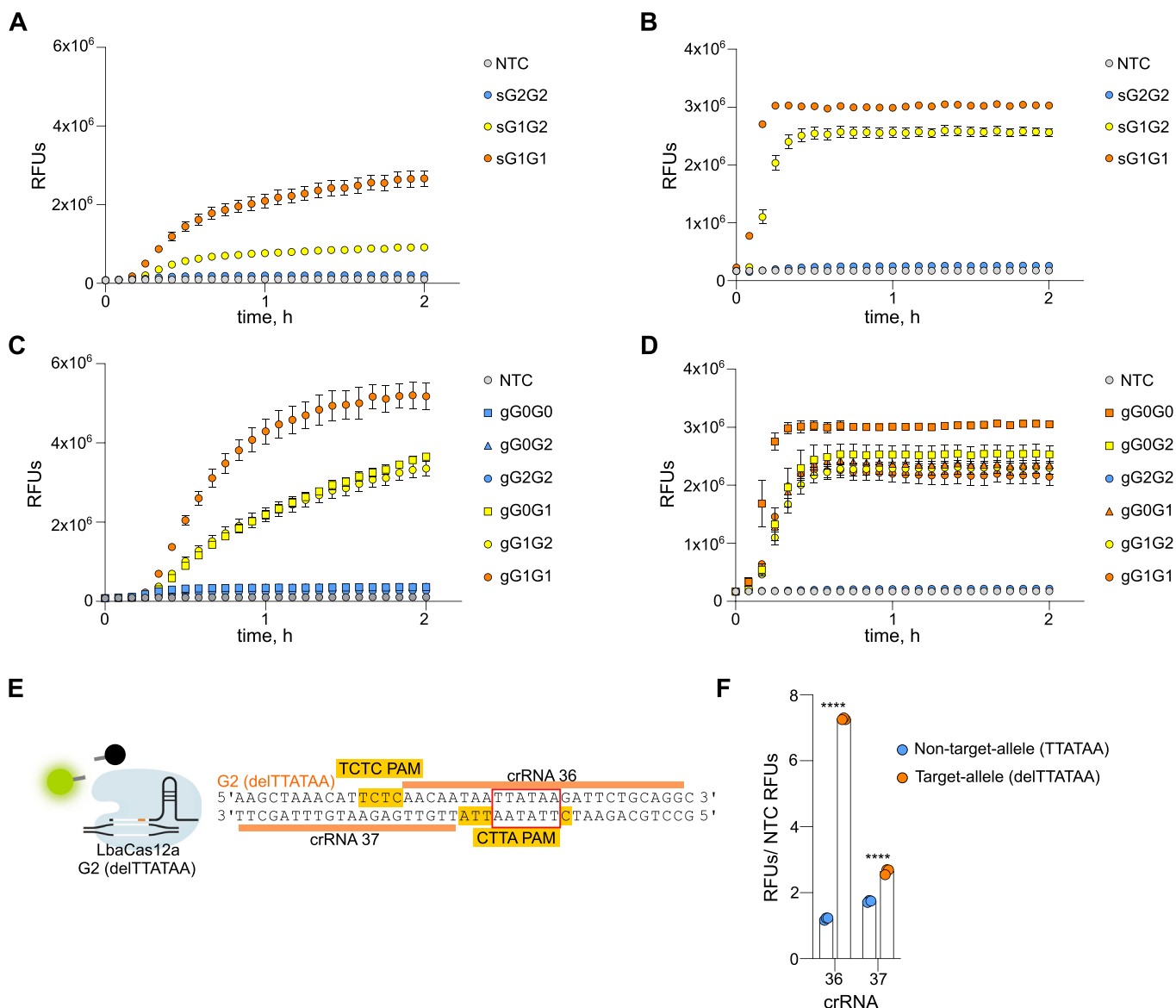

**Figure 3. Comparison of *APOL1* genotyping on synthetic and genomic DNA.**

(A–D) Multiplexed detection of DNA by LwaCas13a (**A**, **C**) and PsmCas13b (**B**, **D**). Synthetic DNA standards were detected at an overall concentration of 100 fM (**A**, **B**) and gDNA was detected at an overall concentration of 10 ng/µl (**C**, **D**). RPA was performed in duplicates for each sample and each RPA product was measured twice with the multiplexed CRISPR assay. Raw RFUs are shown and data are mean ± s.d. (**E**) Alignment of target sequences and G2 mutant sensing crRNAs (LbaCas12a). The spacer region of the crRNAs is shown in orange, and the protospacer adjacent motifs (PAM) are highlighted yellow. Red box highlights the region of the G2 6 base pair deletion. (**F**) crRNA testing for LbaCas12a. CRISPR assay sensing synthetic DNA containing the target allele (orange) and non-target allele (blue). Target DNA was detected at an overall concentration of 300 pM and each data point represents an individual reaction. For each reaction, RFUs are shown at 3 h divided by the mean NTC RFU value. Error bars: s.d. Statistical significance assessed with unpaired *t* test; $P > 0.05$= not significant (ns), *$P ≤ 0.05$, **$P ≤ 0.01$, ***$P ≤ 0.001$ and ****$P ≤ 0.0001$. For *P* values see: Appendix Table S2. Source data are available online for this figure.

detection of the non-target allele sG2G2. We next assessed if the ability to detect and discriminate between the synthetic DNA targets could be extended to gDNA isolated from human blood (Fig. 3C,D). Similarly to synthetic DNA targets, we found that LwaCas13a could correctly discriminate all possible G1 variants (Fig. 3C). In contrast, PsmCas13b did not enable discrimination of wt and heterozygous human gDNA samples (Fig. 3D). Specifically, G2 heterozygous samples with the genotypes G0G2 and G1G2 resulted in a

fluorescence signal indistinguishable from the G2 wt samples G0G1 and G1G1.

To improve G2 genotyping of human gDNA such that wt, heterozygous, and homozygous alleles could be discriminated between, we additionally incorporated LbaCas12a to facilitate the detection of the G2 deletion (mutant allele). Thereby, it would not be necessary for PsmCas13b to distinguish between the target allele and heterozygous alleles regarding the G2 variant. Instead, signal

intensities for both PsmCas13b and LbaCas12a would be classified as either low for non-target allele detection or high for both target allele and heterozygous detection.

Unlike LwaCas13a and PsmCas13b, LbaCas12a relies on a protospacer adjacent motif (PAM) to enable target recognition and subsequent cleavage activity. In addition to its canonical TTTV-PAM, numerous non-canonical PAMs such as TCTV, TTCV, and CTTV have been reported (Chen et al, 2020; Zhou et al, 2022). Since there was no canonical TTTV-PAM in proximity to the G2 deletion, we designed crRNAs that utilize non-canonical PAMs (Fig. 3E) and explored two different strategies for positioning of the G2 deletion: one within the spacer next to a TCTC-PAM (crRNA 36) and the other next to the spacer and within a CTTA-PAM (crRNA 37). crRNA 36 resulted in a greater ability to discriminate between target- and non-target allele synthetic DNA (Fig. 3F) and was selected as G2 mutant sensing crRNA (LbaCas12a) for analysis of human gDNA.

## Multiplexed CRISPR genotyping for detection of two APOL1 variants comprising six genotypes

To multiplex LwaCas13a, PsmCas13b, and LbaCas12a in one assay, we evaluated their orthogonal cleavage activity of six reporter oligonucleotides, differing in sequence and fluorophore-quencher combinations (Fig. EV1). For each reporter molecule, we measured the resulting signal intensity for each enzyme when detecting a target allele standard. We chose commercially available RNase Alert and AAAAA-Texas Red as RNA reporter molecules for LwaCas13a and PsmCas13b, respectively, and we selected TTATT-Hexachlorofluorescein (HEX) as the DNA reporter molecule for LbaCas12a. These choices resulted in a high signal intensity for each corresponding Cas enzyme and a low signal intensity for the other two Cas enzymes.

Utilizing the optimum crRNAs, RPA primers, and reporter molecules, we next tested whether the 6 APOL1 genotypes could be accurately discriminated in the multiplexed assay (Fig. 4A). Genotypes were determined based on signal intensities in the Fam, Texas Red, and Hex channels with a unique combination representing each of the six APOL1 genotypes (Fig. 4B). We first used synthetic DNA mimicking the different genotypes. We found that LwaCas13a (Fig. 4C, upper panel) correctly discriminated all G1 variants showing the highest signal intensity when detecting the target allele (sG1G1), followed by a medium signal intensity when detecting heterozygous alleles (sG0G1, sG1G2), and the lowest signal intensity when detecting the non-target allele (sG0G0, sG0G2, sG2G2). LbaCas12a (Fig. 4C, middle panel) correctly identified G2 deletions, whereby the target allele and heterozygous alleles (sG0G2, sG1G2, sG2G2) resulted in a higher signal intensity as compared to the non-target allele (sG0G0, sG0G1, sG1G1). Vice versa, PsmCas13b (Fig. 4C, lower panel) correctly identified G2 wt sequences with the target allele and heterozygous alleles (sG0G0, sG0G1, sG0G2, sG1G2, sG1G1) resulting in higher signal intensities as compared to the non-target allele (sG2G2).

We next evaluated the ability of the multiplexed assay to distinguish between different APOL1 genotypes using gDNA samples isolated from human blood (Fig. 4D). While the assay accurately identified all human gDNA samples, similar to the results with synthetic DNA, we observed variations in signal intensities across different days, plate readers, and sample

preparations, making result interpretation challenging. To address this, we included target allele DNA standards alongside patient samples for normalization of data. This enabled the calculation of a score for each genotype by dividing the maximum slope of a patient sample, measured between background-subtracted relative fluorescence units (RFUs) from 5 to 60 min, by the maximum slope of the synthetic standard (Fig. 4E).

## Validation of CRISPR genotyping in a clinical cohort

We next assessed our multiplexed CRISPR-genotyping assay by testing 124 patient samples from three cohorts (Germany, USA, Brazil) (Fig. 5). As a reference, blood-derived gDNA was genotyped for the APOL1 variants using PCR followed by Sanger sequencing or quantitative real-time PCR (qPCR). CRISPR-genotyping was performed blinded and genotypes were assigned as described above (Fig. 4B). We next defined genotype score thresholds that allowed us to distinguish wild-type, heterozygous and homozygous G1 and G2 variants (Fig. 5A). As an integrated quality control for successful target amplification and detection, we classified samples with genotype scores below the lowest respective thresholds or that did not result in a signal intensity combination corresponding to an APOL1 genotype as invalid. Overall, our assay achieved an accuracy of 97.9% as compared to the gold standard method (Fig. EV2).

To assess the assay's reproducibility under various conditions, including different sample processing methods, assay reagents, operators, and read-out technologies, we conducted measurements on a subset of the samples at a different clinical center (Fig. EV3). We observed that the genotype score method also resulted in correct discrimination of the six genotypes with high accuracy. However, for LwaCas13a-based G1 detection, the genotype scores were generally higher which required readjustments of the thresholds. To eliminate the need of such adjustments and to further increase the robustness of our analysis, we trained a machine learning model to predict the genotypes based on the signal intensities of the three fluorescent channels (Fig. 5B). Utilizing this model to analyze the data from the 124 samples across both test centers, our assay resulted in an accuracy of 100%, validating its precision for APOL1 genotyping (Fig. 5C).

## Adapting multiplexed CRISPR detection for lateral-flow readout

Finally, we explored if the assay could be optimized toward a lateral-flow-based readout, enabling genotyping at the point-of-care. To enable the simultaneous detection of two analytes, we used lateral-flow sticks that contained both a streptavidin and an anti-digoxigenin test band. We designed two multiplexed assays for G1 and G2 genotyping (Fig. 6A). The assays each contained LwaCas13a-crRNA and LbaCas12a-crRNA complexes, and their respective biotin- and digoxigenin-labeled reporter oligonucleotides. These reporter molecules additionally contained Fam, binding to anti-Fam gold nanoparticles (AuNPs) in the sample application area of the lateral-flow sticks. Upon application of the CRISPR reaction product, reporter molecules, carrying biotin and digoxigenin, travel up the sticks and bind to streptavidin or anti-digoxigenin test bands, respectively. Intact reporter molecules carrying anti-FAM-AuNPs result in a purple color at the test bands, while cleaved reporter molecules carrying anti-FAM-AuNPs travel further along the stick and bind to a secondary anti-

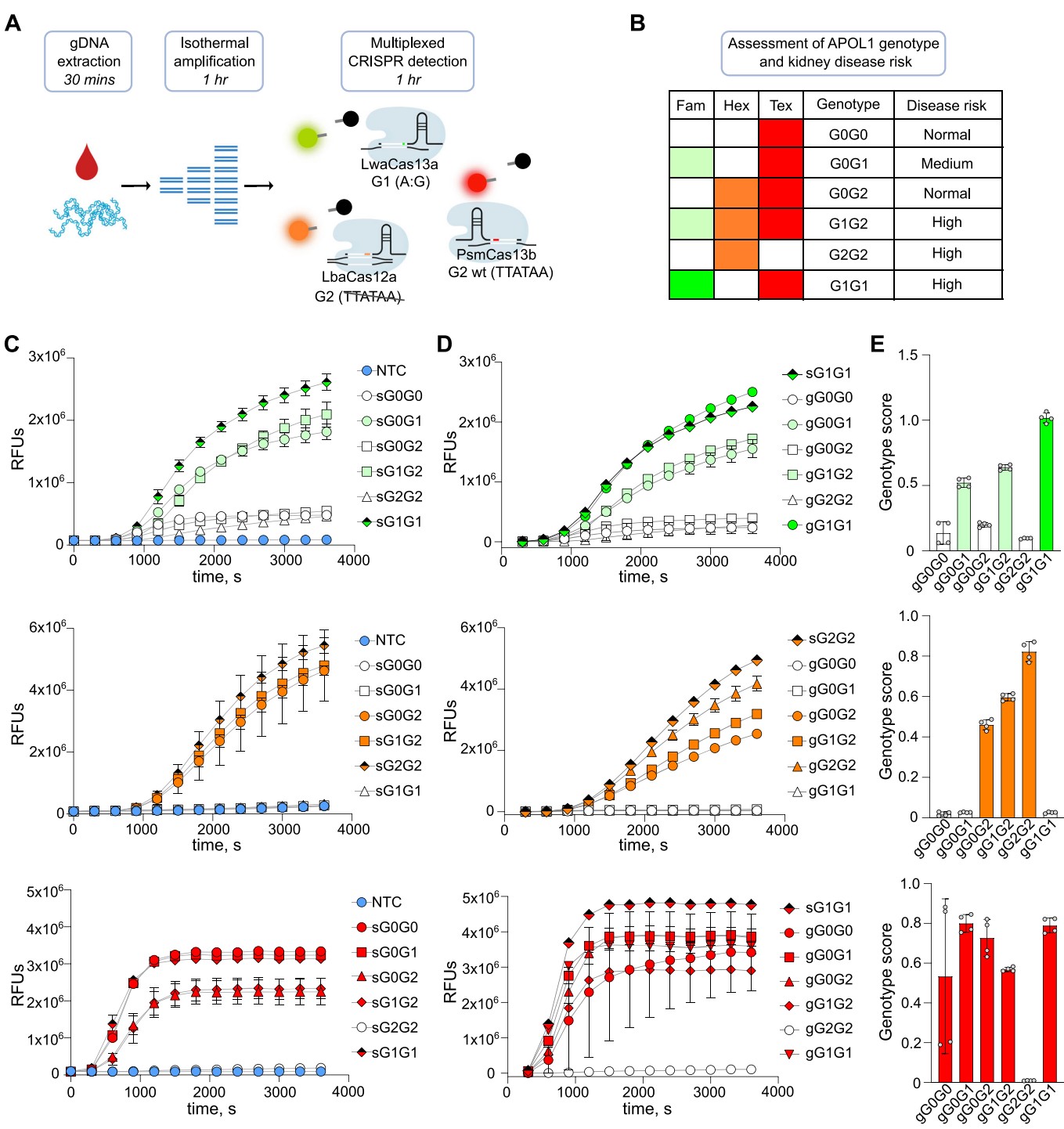

**Figure 4. Multiplexed CRISPR-based detection of six *APOL1* genotypes.**

(A) Schematic illustrating the workflow used to genotype patient samples for the *APOL1* variants. Genomic DNA is amplified by RPA. In a single reaction, amplified DNA is then detected in the multiplexed CRISPR assay directly by LbaCas12a or transcribed to RNA before being detected by LwaCas13a and PsmCas13b. Depending on the Fam, Hex, and Tex signal intensities, the activity of LwaCas13a, LbaCas12a, and PsmCas13b, respectively, is measured. (B) Table summarizing the six unique possible combinations of signal intensities that each determine what *APOL1* genotype is present. (C) Multiplexed detection of DNA standards that mimic the six *APOL1* genotypes. RPA was performed in triplicates and measured once with the CRISPR assay. Raw RFUs are shown and data are mean of three individual reactions. (D) Multiplexed detection of genomic DNA isolated from clinical samples ($n = 6$). Synthetic DNA standard for normalization between experiments (square shaded half black). RPA was performed in duplicates and measured twice with the CRISPR assay. Background-subtracted RFUs are shown. (E) Genotyping scores for data in D ($n = 6$). (C–E) Error bars: s.d. Data for LwaCas13a (upper panel; Fam channel), LbaCas12a (middle panel; Hex Channel), PsmCas13b (lower panel; Texas Red channel). Source data are available online for this figure.

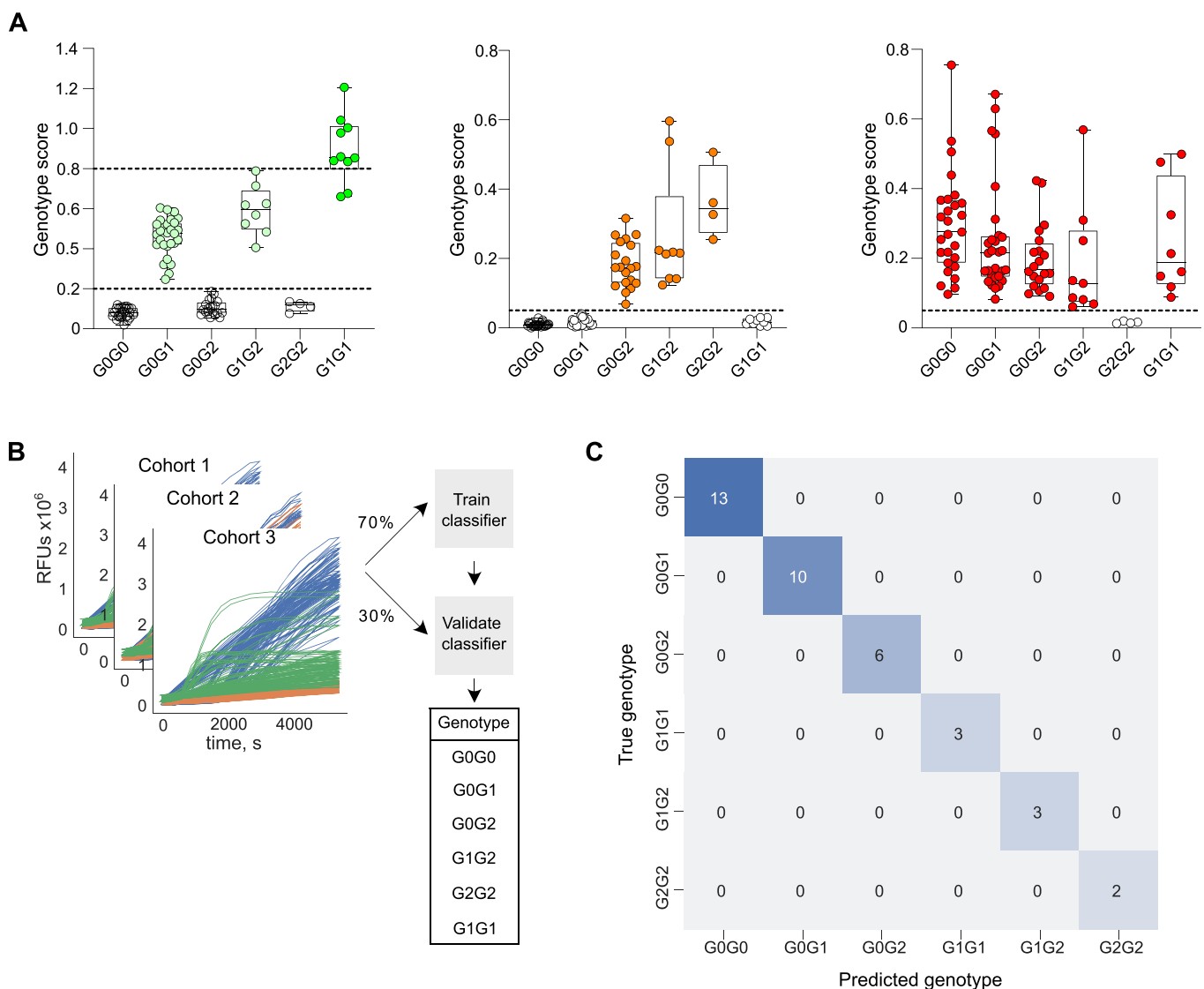

**Figure 5. CRISPR-based *APOL1* genotyping in a multicenter patient cohort.**

(A) Box plots show resulting data of samples from the USA and Germany cohorts that have been genotyped with the multiplexed CRISPR-based assay. The data for LwaCas13a, LbaCas12a, and PsmCas13b are shown on the left, middle, and right, respectively. Dashed lines indicate genotype score thresholds. For LwaCas13a, a score below 0.2 represents wt, a score between 0.2 and 0.8 represents heterozygous, and a score above 0.8 represents homozygous for the G1 SNP. A score below 0.05 for LbaCas12a or PsmCas13b represents wt or homozygous for the G2 deletion, respectively. Data are the mean of six technical replicates with each point representing a different patient. Within each box-whisker plot, whiskers extend from the minimum to the maximum values; boxes extend from the 25th to the 75th percentiles; the median value is annotated by a horizontal line through the box. (B) Schematic of the machine learning model. (C) Confusion matrix summarizing accuracy of CRISPR-based (predicted) genotyping for six *APOL1* genotypes determined by the machine learning model for the USA, Germany, and Brazil cohorts as compared to the true genotypes. Boxes shaded blue indicate a true positive result. Source data are available online for this figure.

species-specific antibody creating a purple control band. We designed the assays such that the absence of a purple test band indicated the positive detection of the target.

For the G1 assay, we included the previously optimized G1 mutant sensing LwaCas13a crRNA (crRNA 5) and a G1 wt sensing LbaCas12a crRNA. To overcome the absence of canonical TTTV PAMs for Cas12a near the G1 SNP, we tested non-canonical C- and T-rich PAMs (Fig. EV4A). We identified a highly discriminative crRNA (crRNA 41) which resulted in a normalized signal intensity that was 4.6 times greater for target allele detection versus

non-target allele detection (Fig. EV4B). For the G2 assay, we used the previously optimized mutant sensing LbaCas12a crRNA (crRNA 36) and tested different G2 wt sensing LwaCas13a crRNAs, selecting a crRNA (crRNA 62) with a target:non-target allele signal intensity ratio of 26.8. (Fig. EV4C,D).

Subsequently, we tested the ability to discriminate the six different *APOL1* genotypes for both synthetic and genomic DNA (Fig. 6B). Regarding both the G1 and G2 variants, we observed three distinct patterns on the respective lateral-flow sticks for both synthetic and genomic DNA: wt samples (G0G0, G0G2, G2G2 for

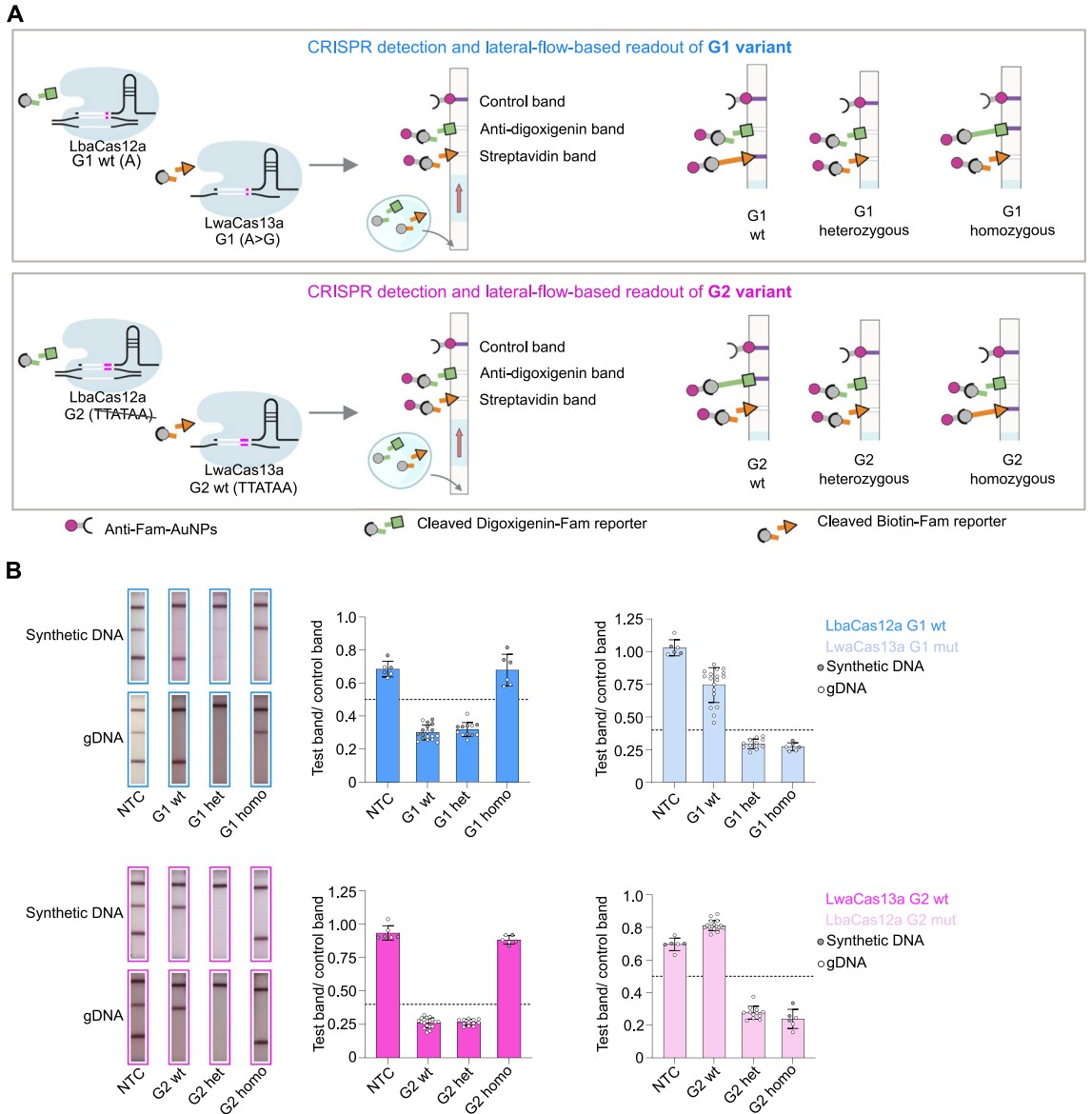

**Figure 6. Multi-analyte lateral-flow readout of multiplexed CRISPR genotyping.**

(A) Schematic illustrating the lateral-flow-based assay. Multiplexed CRISPR detection by LbaCas12a and LwaCas13a for *APOL1* alleles G1 (top) and G2 (bottom). Upon target recognition, the collateral cleavage of reporter molecules results in the absence of gold nanoparticles at the corresponding test band. (B) Lateral-flow-based readout of *APOL1* genotyping. Representative images for genotypes wt, het, or homo for variants G1 (highlighted blue) and G2 (highlighted pink) shown for synthetic DNA and patient samples. Quantification of the test band to control band intensity ratios for patient samples (white) and synthetic DNA (gray) is displayed as the mean ± s.d. RPAs performed in triplicates for each genotype and each RPA product measured once with multiplexed G1 and G2 assays; each white point represents one patient and each gray point represents an individual reaction. The dashed line indicates a relative band intensity of 0.4 (LwaCas13a) and 0.5 (LbaCas12a). Source data are available online for this figure.

the G1 assay; G0G0, G0G1, G1G1 for the G2 assay) resulted in the absence of the test band corresponding to wt-detecting Cas-crRNA complexes; heterozygous samples (G0G1, G1G2 for the G1 assay; G0G2, G1G2 for the G2 assay) resulted in the absence of both test bands; homozygous mutant samples (G1G1 for the G1 assay; G2G2 for the G2 assay) resulted in the absence of the test band corresponding to mutant-detecting Cas-crRNA complexes. When considering both the G1 and G2 assays, each of the six *APOL1* genotypes thereby produce a unique test band pattern observable

visually that enables accurate *APOL1* genotyping without the need for additional equipment (Fig. 6B).

# Discussion

*APOL1*-mediated kidney disease is now recognized as one of the most common genetic kidney diseases in humans. Fast and reliable genotyping is crucial not only for disease diagnosis but also for use

in transplantation for risk-stratifying kidney donor organs prior to allocation. Current available genotyping methods rely on PCR and sequencing and are associated with high equipment costs, long turnaround times and bioinformatic data analysis, all of which can challenge their application in resource-limited settings or at the point-of-care. Further, hybridization-based genotyping only allows for genotyping of few variants per reaction limiting multiplexing capability. CRISPR-based genotyping has the potential to complement current genetic diagnostics through its speed, specificity and ease of use, which makes it compatible for point-of-care testing.

Recent studies in CRISPR-based SNP detection have made significant progress in utilizing the single-base specificity of Cas effectors for discriminating various targets, such as virus subtypes (Myhrvold et al, 2018; De Puig et al, 2021), cancer mutations (Gootenberg et al, 2018; Teng et al, 2019), pharmacogenetic variants (Wu et al, 2023; Chen et al, 2021), genetic diseases (Azhar et al, 2021), and SNPs associated with disease risks (Li et al, 2018). However, these studies predominantly rely on PCR-based preamplification methods (Kumar et al, 2021; Liang et al, 2021; He et al, 2022; Welch et al, 2022; Harrington et al, 2018), have rarely been evaluated with extensive clinical samples (Azhar et al, 2021; Balderston et al, 2021), and, to our knowledge, have not achieved multiplexed detection necessary for detecting more than one variant.

In this study, we developed a multiplexed CRISPR-based assay that detects two genetic variants that occur in the *APOL1* gene, termed G1 and G2. By introducing synthetic mismatches between the target and spacer sequence, we fine-tuned the cleavage activity of LwaCas13a to enable discrimination between samples wt, heterozygous, or homozygous for the G1 SNP. Specifically, this was achieved by incorporating a second synthetic mismatch, differing from current crRNA design approaches and highlighting the need for optimization of the crRNA for each target (Gootenberg et al, 2018).

In addition, we optimized the cleavage activity of both LbaCas12a and PsmCas13b to enable their collective discrimination for the G2 deletion. Our resulting assay advances CRISPR diagnostics by enabling the genotyping for two variants in one reaction, thus discriminating between the subsequent six possible genotypes.

We validated the assay's performance through genotyping a multicenter clinical cohort of over 100 patients and by establishing a machine learning-based analysis method that enables a robust readout for fluorescence-based genotyping even with different conditions such as machines, personnel and reagent batches. The accuracy achieved was comparable to the current clinical gold standard method and, to our knowledge, this is the largest cohort used in a CRISPR-based genotyping study to date.

We further demonstrated for the first time a multi-analyte lateral-flow-based readout of multiplexed CRISPR genotyping, advancing its potential for point-of-care applications. The assay was validated on clinical samples representing all six *APOL1* genotypes.

While these advancements are important for CRISPR genotyping and are crucial for clinical use and point-of-care compatibility, remaining challenges include PAM restrictions of Cas enzymes that limit their potential targets, reliance on orthogonal cleavage preferences and fluorophores for multiplexing, and the lack of a comprehensive database or tools to enable the choice of Cas enzymes, crRNAs and reporter molecules without the need for prior experimental screening.

In clinical practice, ethnicity has been used as a risk factor when assessing kidney disease. In addition, the kidney donor profile index (KDPI) calculator incorporates ethnicity as a variable to predict graft loss risk in kidney transplantation. However, recent studies have demonstrated that the identification of donors carrying two high-risk *APOL1* variants, rather than relying on ethnicity, would significantly enhance the risk prediction capability of the KDPI (Julian et al, 2017).

Recurrence of kidney disease is a leading cause of transplant loss (Uffing et al, 2021), and *APOL1* testing would help guide counseling of patients about their risk of kidney disease recurrence post-transplantation. It could also play a role in evaluating potential living kidney donors, as the presence of two high-risk *APOL1* variants may impact donation recommendations. The allocation of organs based on donor *APOL1* status is currently under investigation by the *APOL1* Long-term Kidney Transplantation Outcomes Network Study (Freedman et al, 2020). While patient advocacy groups have indicated support of *APOL1* genotyping (Umeukeje et al, 2019; Young et al, 2019), sequencing-based approaches are challenged by high costs and slow turnaround time.

Currently, there are no approved disease-specific therapies for *APOL1*-associated nephropathy. However, recent early-stage clinical trials of *APOL1* inhibitors have emerged as a promising treatment option for *APOL1*-mediated kidney disease. The small-molecule drug inaxaplin inhibits *APOL1*-mediated pore formation in podocyte membranes and prevents excess ion flux and osmolysis (Egbuna et al, 2023). In a Phase 2a trial, inaxaplin reduced proteinuria in patients carrying two *APOL1* risk alleles with FSGS. Rapid, low-cost genotyping of *APOL1* risk alleles is crucial for guiding targeted treatments.

In conclusion, CRISPR-based genotyping may help making accurate diagnosis of *APOL1*-mediated kidney disease widely accessible due to its increased speed and cost-effectiveness as compared to Sanger sequencing, thereby enabling early preventative measures and monitoring of disease (Appendix Table S3). Further, it could serve as companion diagnostics for future genotype-guided treatments and finally provide genetic information in the point-of-care setting such as organ transplantation. Therefore, CRISPR-based genotyping has the potential to advance precision medicine approaches in the field of nephrology.

## Methods

**Reagents and tools table**

| Reagent/resource | Reference or source | Identifier or catalog number |
|---|---|---|
| **Experimental models** | | |
| NA | | |
| **Recombinant DNA** | | |
| pC013 | Addgene | 90097 |
| pC0061 | Addgene | 115211 |
| **Antibodies** | | |
| NA | | |
| **Oligonucleotides and other sequence-based reagents** | | |

| Reagent/resource | Reference or source | Identifier or catalog number |
|---|---|---|
| RPA primers | This study | Appendix Table S1 |
| crRNA constructs | This study | Appendix Table S1 |
| Synthetic RNA target constructs | This study | Appendix Table S1 |
| Synthetic DNA targets | This study | Appendix Table S1 |
| Reporter molecules | This study | Appendix Table S1 |
| PCR primers | This study | Appendix Table S1 |
| **Chemicals, enzymes, and other reagents** | | |
| RNase Alert QC systen | ThermoFisher Scientific | 4479769 |
| Murine RNase inhibitor | New England Biolabs | M0314L |
| Buffer 2.1 | New England Biolabs | B7202S |
| Ribonucleotide solution set | New England Biolabs | N0450L |
| NxGen T7 RNA polymerase | Lucigen | 30223-2-LU |
| LwaCas13a | Gootenberg et al, 2017 | |
| LbaCas12a | New England Biolabs | M0653T |
| PsmCas13b | Gootenberg et al, 2018 | |
| **Software** | | |
| GraphPad Prism 8.4.3 | https://www.graphpad.com/ | |
| ImageJ (National Institutes of Health) | https://imagej.net/ij/ | |
| **Other** | | |
| DNeasy Blood & Tissue kit | Qiagen | 69504 |
| Hybridetect 2 T lateral-flow sticks | Milenia Biotec | MGHD2 1 |
| Spectramax multi-mode microplate reader iD3/iD5 | Molecular Devices | 735-0391/ 76175-288 |
| Heat block | Eppendorf | E-5055 |
| 384-well microplate | Corning | CLS3544-50EA |
| HiScribe T7 Quick High Yield RNA Synthesis kit | New England Biolabs | E2050S |
| Monarch RNA Cleanup Kit | New England Biolabs | T2050L |
| RNA Clean & Concentrator 25 | Zymo | 13175 |
| TwistAmp Basic RPA kit | Twistdx | TABAS03KIT |
| Multienzyme isothermal rapid amplification kit | AmpFuture biotech | WLB8201KIT |
| Centrifuge | Eppendorf | E-0812 |
| Thermal cycler | Eppendorf | 6311000010 |

## Methods and protocols

### LwaCas13a and PsmCas13b protein purification

The expression vectors pC013-Twinstrep-SUMO-huLwaCas13a (Addgene plasmid #9009, RRID: Addgene_90097 (Gootenberg et al, 2017)) and pC0061 PsmCas13 (B05) His6-TwinStrep-SUMO-BsaI (Addgene plasmid #115211, RRID:Addgene_115211 (Gootenberg et al, 2018)) were a gift from Feng Zhang. The Cas13a protein

from *Leptotrichia wadeii* and the Cas13b protein from *Prevotella sp.* MA2016 were produced using *E. coli* T7 Express cells (NEB) co-transformed with the pRARE2 plasmid. TB media was supplemented with 100 µg/mL ampicillin and 34 µg/mL chloramphenicol. The cultures were grown at 37 °C until the $OD_{600}$ reached about 2. Gene expression was induced by the addition of 0.5 mM isopropyl β-D-1-thiogalactopyranoside at 17 °C. After induction, cultures were grown overnight at 17 °C. Cells were harvested by centrifugation and the pellets were stored at −80 °C.

For purification, cells were resuspended in lysis buffer (50 mM Tris pH 8.0, 0.5 M KCl, 5% glycerol, 1 mM $MgCl_2$), supplemented with cOmplete™ (EDTA-free protease inhibitor cocktail, Roche), 0.5 mM dithiothreitol (DTT), 1 mM phenylmethyl-sulfonyl fluoride, and 6.000 U/mL lysozyme (Serva), and lysed by sonication (SONOPULS HD 2200, Bandelin Electronic GmbH & Co. KG). The extract was cleared by centrifugation at 36.000 × g and the His6-fusion protein was captured from the supernatant using affinity chromatography on a 5 mL Ni Sepharose® 6Fast Flow column (Cytiva), equilibrated with 50 mM Tris-HCl pH 8.0, 0.5 M KCl, 5% glycerol, 1 mM $MgCl_2$, 5 mM imidazole pH 8.0, and 0.5 mM DTT. Before elution with elution buffer (20 mM Tris-HCl pH 8.0, 0.5 M KCl, 5% glycerol, 0.5 mM DTT, 0.2 M imidazole pH 8.0), the bound protein was washed three times with various wash buffers (1: 20 mM Tris-HCl pH 8.0, 1 M KCl, 5% glycerol, 0.5 mM DTT, 5 mM imidazole pH 8.0; 2: 20 mM Tris-HCl pH 8.0, 0.5 M KCl, 5% glycerol, 0.5 mM DTT, 5 mM imidazole pH 8.0; 3: 20 mM Tris-HCl pH 8.0, 0.5 M KCl, 5% glycerol, 0.5 mM DTT, 15 mM imidazole pH 8.0) to remove contaminating nucleic acids and proteins. The eluted protein was supplemented with 5% glycerol and 0.5 mM DTT, and the fusion tag was cleaved off by adding 1:75 (w/w) yeast Ulp1p SUMO protease (produced in-house). The protein was further purified by ion exchange chromatography on a HiTrap Heparin HP column (Cytiva), equilibrated with 20 mM Hepes-NaOH pH 7.5, 0.25 M KCl, 5% glycerol, and 1 mM DTT. The pooled fractions were supplemented with 5 mM $MgCl_2$, concentrated and applied onto a 26/600 Superdex 200 prep grade column (Cytiva) equilibrated with 50 mM Tris-HCl pH 7.5, 1 M NaCl, and 1 mM DTT. The storage buffer of the protein was adjusted to 50 mM Tris-HCl pH 7.5, 600 mM NaCl, 5% glycerol, and 2 mM DTT. The purified protein was flash-frozen in small aliquots with liquid nitrogen and stored at −80 °C until further use.

### Sample preparation

Patient samples of the Brazil and USA cohorts were prepared using the DNeasy Blood & Tissue kit (Qiagen), according to the manufacturer's instructions. Patient samples of the German cohort were prepared using either the Gentra Puregene Kit (Qiagen) or using magnetic beads and the QIAsymphony SP machine (Qiagen), according to the manufacturer's instructions. After DNA extraction and purification, the DNA integrity was assessed by their 260/280 ratio with a spectrophotometer (Nano-drop, ThermoFisher). Samples were aliquoted and stored at −80 °C.

The samples of the Brazil cohort were genotyped for the *APOL1* G1 and G2 variants using qPCR (Riella et al, 2019). For this, samples were diluted to 20 ng/µl. The qPCR assay consisted of two reactions, with two sets of TaqMan probes in each reaction. The G1 assay has 1 probe complementary to the G1 site and another complementary to the G0 sequence at the same single nucleotide variant coordinate. Similarly, for the G2 assay, 1 probe detects the

G2 sequence, and another detects the G0 sequence at the same coordinates of the 6 base pair deletion. Each probe is tagged with a different fluorophore. Each assay plate run included human positive controls for the combination of genotypes: G0/G0, G0/G1, G0/G2, G1/G1, G2/G1, G2/G2, assayed with both G1 and G2 TaqMan probes. Amplification results were interpreted with the QuantStudio Real-Time PCR Software V1.2 (Applied Biosystems, Foster City, CA).

The samples of the USA and Germany cohorts were genotyped by Sanger Sequencing (Eurofins). Samples were first amplified by PCR. The PCR products were then purified using a PCR purification kit as per the manufacturer's instructions (Qiagen).

### RPA primers and reaction

Primers were designed using NCBI Primer-BLAST, with lengths of 25–30 nt and a melting temperature of 57–67 °C. Combinations of primers were tested for their ability to amplify a synthetic DNA target resulting in amplicons that include both the G1 and G2 coordinates. The primers were designed such that the forward primer is at the 5' end of the G1 and G2 coordinates, while the reverse primer is at their 3' end. Forward primers contained a T7 promoter overhang so that amplified DNA could be converted to RNA by T7 transcription.

RPA reactions were completed for 1 h at 39 °C at a total volume of 20 μl, including 2 μl DNA as a sample input, using either the TwistAmp Basic kit (TwistDx) or the multienzyme isothermal rapid amplification kit (AmpFuture biotech). RPA using the TwistAmp Basic kit was completed with the following modifications: per reaction, 0.267x of a pellet was used with 120 nM forward primer, 480 nM reverse primer, 11.8 μl rehydration buffer, 8 mM MgAOc. RPA using the multienzyme isothermal rapid amplification kit was completed with the following modifications: per reaction, 0.4× of a pellet was used with 120 nM forward primer, 480 nM reverse primer, 12.48 μl Buffer A, 1 μl Buffer B.

### crRNA design

crRNA constructs were designed so that synthesized LwaCas13a, PsmCas13b, and LbaCas12a crRNAs included spacers with a length of 28, 30, and 20 nucleotides, respectively, and are reverse complement to the target site. DNA oligos (Eurofins) with a T7 promoter sequence were used as constructs for in-vitro RNA transcription of the crRNAs. LwaCas13a and LbaCas12a constructs contained a direct repeat sequence at the 5' end of the spacer, while PsmCas13b constructs contained a direct repeat at the 3' end of the spacer.

### Reporter molecules

For fluorescence-based readout, reporter molecules included sequences that were collaterally cleaved by each Cas enzyme (AU or UUUUU for LwaCas13a; AAAAA for PsmCas13b; TTATT for LbaCas12a). Reporter molecules included a quenched fluorophore, consisting of Texas red, Hex, or Fam and were ordered as RNA or ssDNA (IDT). For lateral-flow-based readout, reporter molecules included Fam and biotin (LwaCas13a) or Fam and digoxigenin (LbaCas12a).

### Production of crRNAs and target RNAs

Synthesis of RNA was completed using the HiScribe T7 Quick High Yield RNA Synthesis kit (New England Biolabs) according to the manufacturer's instructions and purified using either the Monarch RNA Cleanup Kit (50 μg, New England Biolabs) or the RNA Clean & Concentrator 25 (Zymo). RNA purity was measured using a spectrophotometer (Nano-drop; Thermofischer).

### Fluorescence-based detection assay

In all, 3 μL of RPA product was detected in a 20 μL CRISPR reaction in a 384-well microplate at 37 °C. CRISPR detection was performed with final concentrations of 250 nM poly-A-Texas Red reporter, 250 nM AU-Texas red reporter, 125 nM RNase Alert, 40 U/μl murine RNase inhibitor (NEB), 1× NEB buffer 2.1, 4 mM rNTPs, 1.5 U/μl T7 RNA polymerase. Cas enzyme:crRNA ratios were kept at ratios of 2:1. Fluorescence was measured on a multi-mode microplate reader; iD5 (Germany and USA cohorts) or iD3 (Brazil cohort) for three different excitation/emission wavelength pairs (485/ 525 nm for Fam; 530/570 nm for Hex, and 585/625 nm for Texas red). Fluorescence measurements were read for at least 1 h at 5-min intervals.

### Calculation of genotype scores

For calculating the genotype score when detecting gDNA, the slope of the resulting background-subtracted fluorescence data (RFUs) was calculated for consecutive fluorescence measurements between 5 and 60 min. The maximum slope for each replicate was then normalized to the mean of the maximum slopes of all replicates for a synthetic standard containing the target allele (sG1G1 for LwaCas13a and PsmCas13b, and sG2G2 for LbaCas12a).

### Machine learning model

We performed multivariate time-series classification using the sktime library (Löning et al, 2019). One multivariate time trace consists of time traces for the three enzymes, and traces were split randomly into 70% training and 30% validation set per genotype. We used a Canonical Interval Forest with 40 estimators (Middlehurst et al, 2020), and estimated the accuracy of the classifier by doing 100 random splits into training and validation sets and then taking the average classification accuracy. We did not observe increased accuracy when performing data augmentation or class balancing (Nikitin et al, 2024). Code is included in the SourceData_Code folder.

### Lateral-flow-based detection assay

Lateral-flow-based detection was performed as described for the fluorescence-based assay, with the following modifications. G1 detection reactions were performed with 90 nM LwaCas13a, 360 nM LbaCas12a, 45 nM LwaCas13a crRNA, 180 nM LbaCas12a crRNA, 2 mM rNTPs, 5 or 2.5 nM for biotin-Fam reporter (for synthetic or genomic DNA, respectively) and 5 nM digoxigenin-Fam reporter. G2 detection reactions were performed with 90 nM LwaCas13a, 180 nM LbaCas12a, 45 nM LwaCas13a crRNA, 90 nM LbaCas12a crRNA, 4 or 2 mM rNTPs (for synthetic or genomic DNA, respectively), 5 nM for each Biotin-Fam reporter and Digoxigenin-Fam reporter. RPA was performed for 1 h, followed by T7-based RNA transcription and CRISPR detection for 1 h at 37 °C. Overall, 20 μl CRISPR reactions were mixed with 80 μl assay buffer followed by insertion of the lateral-flow sticks (Hybridetect 2 T, Milenia Biotec) and incubation for 10 min at room temperature before images were taken.

### The paper explained

#### Problem

Detecting genetic variants enables the identification of disease risk factors and the initiation of preventative treatments. However, current genotyping methods largely rely on sequencing and PCR, limiting their accessibility at the point of care.

Here, we report a CRISPR-based genotyping assay and demonstrate its use in detecting two genetic variants in the Apolipoprotein L1 gene (*APOL1*), which are common among individuals of sub-Saharan African ancestry and greatly increase the risk of kidney disease.

Knowing a patient's genotype can guide renal-protective lifestyle changes. In addition, *APOL1* inhibitors are currently being investigated in clinical trials as a promising genotype-guided treatment option. In kidney transplantation, individuals receiving an organ from a donor with a high-risk *APOL1* variant have worse outcomes and a higher risk of transplant failure. Despite this, commercially available tests take multiple days to deliver results, making timely genotyping for these groups impractical.

#### Results

Using CRISPR-based diagnostics, which offer high sensitivity and specificity for detecting nucleic acids, we report a multiplexed genotyping assay that employs LwaCas13a, PsmCas13b, and LbaCas12a for the simultaneous detection of six *APOL1* genotypes.

Machine learning-assisted analysis of fluorescence-based CRISPR readouts enabled robust and accurate genotype identification across a multicenter clinical cohort of over 100 patients.

In addition, we extended the readout to a point-of-care compatible format using a multi-analyte lateral-flow assay, demonstrating the ability to rapidly determine genotypes from clinical samples with minimal equipment.

#### Impact

CRISPR-based diagnostics offer rapid and accessible identification of genetic risk carriers, enhancing disease risk awareness and supporting the genotype-guided prescription of drugs like APOL1 inhibitors. Its speed and ease of use enable genotyping in emergency situations, during kidney transplant stratification, and in resource-limited settings. Accurate diagnosis of APOL1-mediated kidney disease may help advance precision medicine in nephrology.

### Image analysis of lateral-flow reactions

Using ImageJ software (National Institutes of Health), the relative band intensity was calculated. Image brightness was first adjusted such that the blank part of the lateral-flow stick had a mean gray value of 210. Images were next converted to 8-bit and inverted, and then the region of the band was selected and its mean gray value was measured. For G1 mutant and G1 wt detection, the relative band intensities were calculated as the streptavidin or anti-digoxigenin bands, respectively, divided by the control bands. For G2 mutant and G2 wt detection, the relative band intensities were calculated as the anti-digoxigenin or streptavidin bands, respectively, divided by the control band.

### Patient populations and ethics

Clinical samples from different sites were obtained, and informed consent was obtained from all human subjects. Experiments conformed to the principles set out in the World Medical Association Declaration of Helsinki and the Department of Health and Human Services Belmont Report.

In Brazil, institutional review board approval was obtained from the National Committee for Ethics in Research (Brasília, Brazil). Written informed consent was acquired from all participants. Blood samples were collected in EDTA tubes and kept at 20 °C until DNA extraction.

Additional samples were obtained from discarded material from adults (>18 years) presenting to Massachusetts General Hospital. The material was excess to clinical needs, and selected based on having self-identified ethnicity documented in the Electronic Health Record as "Black or African American" and an estimated glomerular filtration rate (eGFR, as calculated with the CKD-EPI refit 2021 creatinine-based equation) below 45 mL/min/1.73 m². Whole blood samples were frozen at −80 °C before research use. The study was granted exemption from informed consent due to the use of anonymized discarded clinical samples and was approved by the Mass General Brigham IRB, Protocol no. 2022P000747.

Samples were also collected from patients presenting to the Neurology department at the University Hospital rechts der Isar (Technical University Munich (TUM)) after informed consent, and stored in their Biobank which operates under 9/15-S at the local ethics committee (Ethikkommission der Technischen Universität München).

### Statistical analysis

CRISPR-based genotyping of clinical samples was performed blinded. For multivariate time-series classification, traces were split randomly into training and validation sets. All experiments in this study were completed with at least three technical replicates. Statistical analyses were performed using GraphPad Prism 8.4.3 software.

## Data availability

This study includes no data deposited in external repositories.

The source data of this paper are collected in the following database record: biostudies:S-SCDT-10_1038-S44321-024-00126-x.

## Peer review information

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

## Acknowledgements

MMK was supported by the Emmy Noether Programme (grant no. KA5060/1-1) of the German Research Foundation (DFG) and is a participant in the BIH Charité Clinician Scientist Program funded by the Charité— Universitätsmedizin Berlin and the Berlin Institute of Health at Charité (BIH). AJS was supported by the BIH MD Research Stipend. Nora Diederich was supported by the Peter-Scriba Stipend of the German Society of Internal Medicine (DGIM). The authors thank the team of the Protein Production & Characterization Technology Platform of the Max Delbrück Center for Molecular Medicine in the Helmholtz Association (MDC), Berlin, Germany (https://www.mdc-berlin.de/protein-production-characterization) for excellent technical assistance.

## Author contributions

**Robert Greensmith**: Conceptualization; Data curation; Formal analysis; Validation; Investigation; Visualization; Methodology; Writing—original draft; Writing—review and editing. **Isadora T Lape**: Data curation; Investigation. **Cristian V Riella**: Conceptualization; Methodology; Writing—original draft. **Alexander J Schubert**: Data curation; Formal analysis; Investigation. **Jakob J Metzger**: Software; Formal analysis; Visualization; Methodology. **Anand S Dighe**: Resources. **Xiao Tan**: Resources. **Bernhard Hemmer**: Resources. **Josefine Rau**: Investigation. **Sarah Wendlinger**: Visualization; Writing—review and editing. **Nora Diederich**: Visualization; Writing—review and editing. **Anja Schütz**: Resources. **Leonardo V Riella**: Conceptualization; Supervision; Methodology; Writing—original draft. **Michael M Kaminski**: Conceptualization; Supervision; Funding acquisition; Validation; Methodology; Writing—original draft; Project administration; Writing—review and editing.

Source data underlying figure panels in this paper may have individual authorship assigned. Where available, figure panel/source data authorship is listed in the following database record: biostudies:S-SCDT-10_1038-S44321-024-00126-x.

## Funding

## Disclosure and competing interests statement

RG, MMK, ITL, CVR, and LVR have filed a patent application covering the techniques as described in the manuscript. The remaining authors declare no competing interests.

# Expanded View Figures

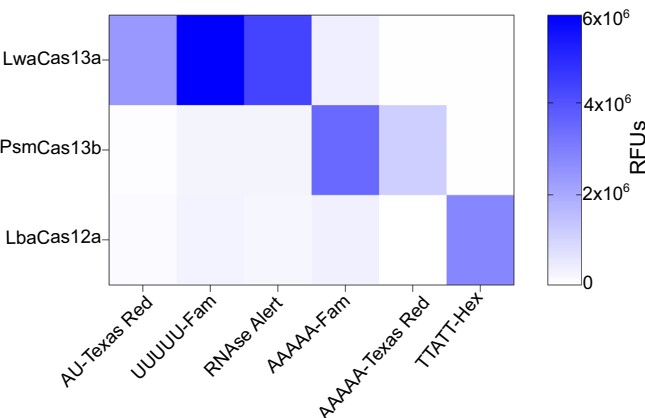

**Figure EV1. Orthogonality of Cas enzyme cleavage preference.**

CRISPR assay sensing synthetic RNA containing the target allele at an overall concentration of 15 nM (LwaCas13a; PsmCas13b) or DNA at an overall concentration of 300 pM (LbaCas12a). RFUs are shown at 3 h and heatmaps visualize the mean of 3 technical replicates. Source data are available online for this figure.

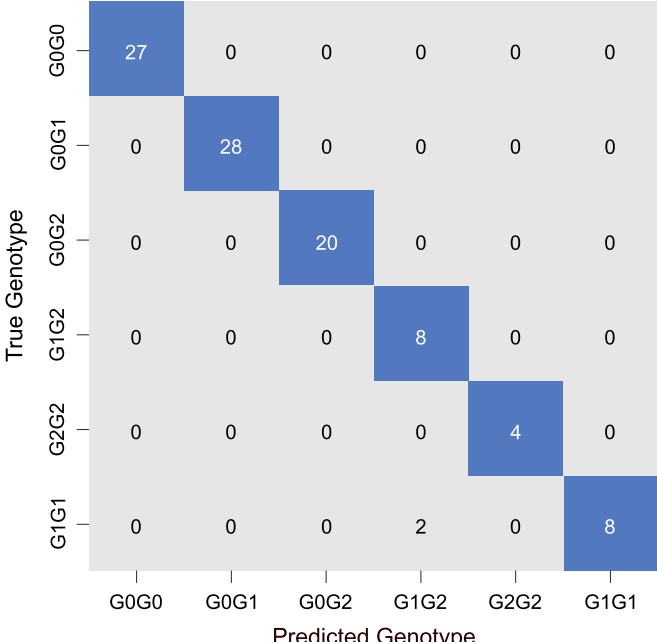

**Figure EV2. CRISPR-based *APOL1* genotyping by genotype scores.**

Confusion matrix summarizing accuracy of CRISPR-based (predicted) genotyping as calculated by the genotype score method for six *APOL1* genotypes as compared to Sanger Sequencing determined (true) genotypes for US and German cohort as shown in Fig. 5A. Numbers indicate patients and boxes shaded blue indicate a true positive result. Source data are available online for this figure.

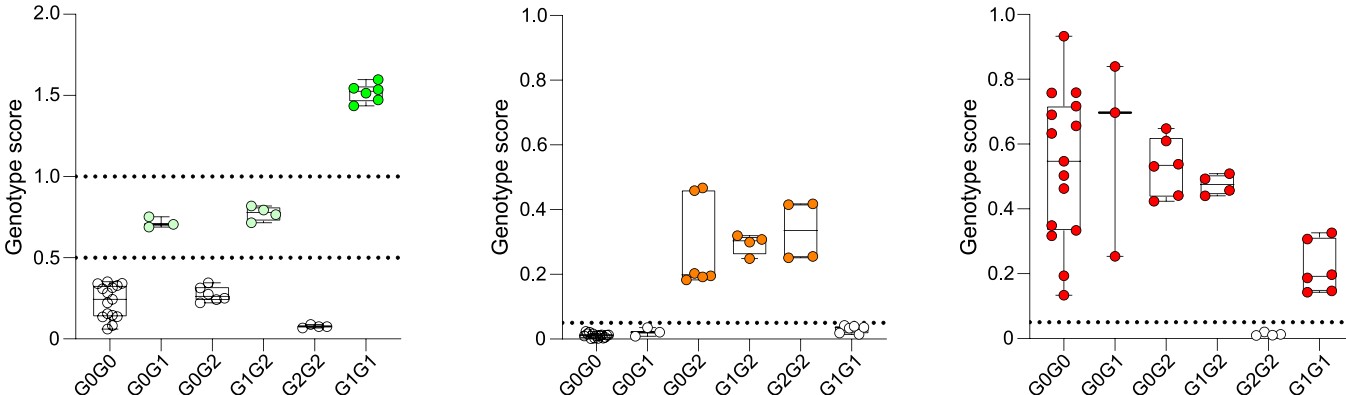

**Figure EV3.   *APOL1* genotyping of independent Brazilian cohort.**

Box plots show genotype scores of patient samples from the Brazilian cohort. The data for LwaCas13a, LbaCas12a, and PsmCas13b are shown on the left, middle, and right, respectively. Each data point represents an individual patient. Dashed lines indicate genotype score thresholds. Within each box-whisker plot, whiskers extend from the minimum to the maximum values; boxes extend from the 25th to the 75th percentiles; the median value is annotated by a horizontal line through the box. Source data are available online for this figure.

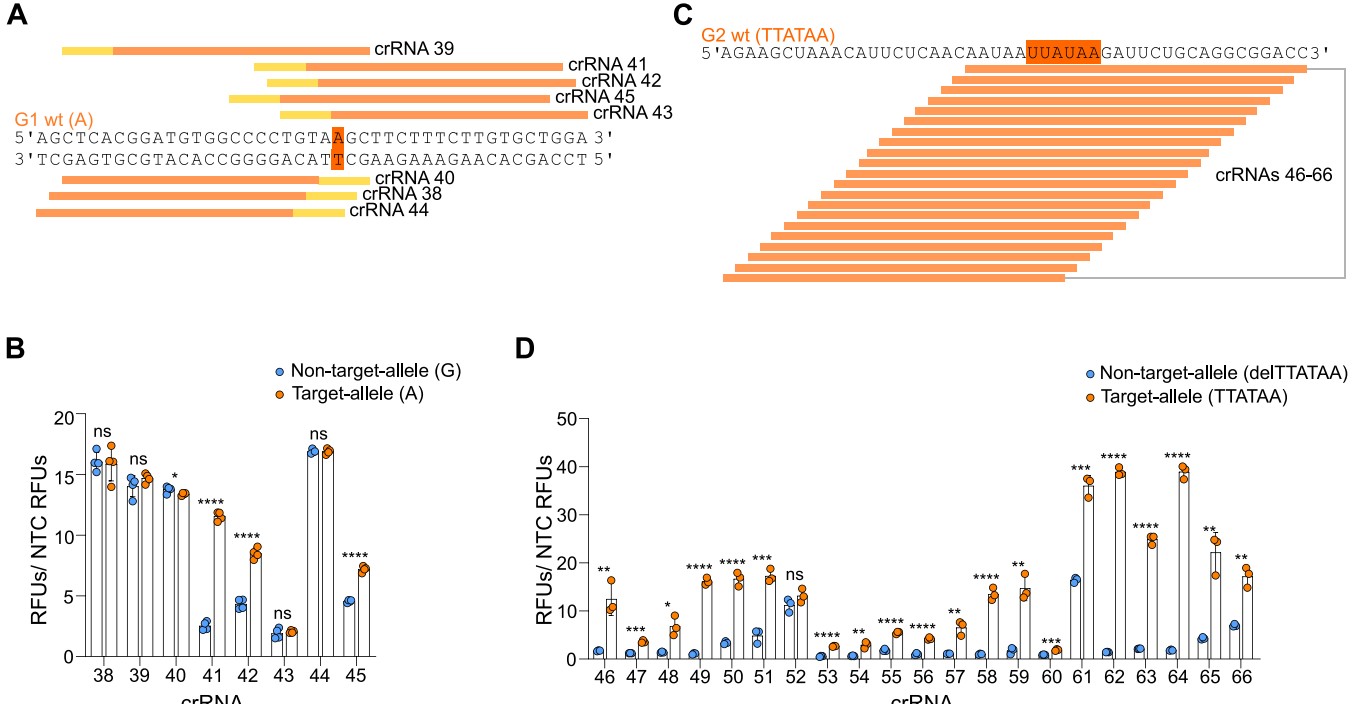

**Figure EV4.   crRNA screening for lateral-flow-based _APOL1_ genotyping.**

(**A**) Alignment of target sequences and LbaCas12a G1 wt sensing crRNAs. The spacers of the crRNAs are shown in orange, and the corresponding PAMs are shown in yellow. G1 wt allele highlighted in dark orange. (**B**) Screen for G1 wt sensing LbaCas12a crRNAs detecting synthetic DNA containing target- (wt; orange) and non-target- (mut; blue) alleles. DNA was detected at an overall concentration of 100 fM. RPA was performed in duplicates for each sample and each RPA product was measured twice with the CRISPR assay. Each data point represents an individual reaction. For each reaction, relative RFUs are shown at 2 h divided by the mean NTC RFU value. (**C**) Alignment of target sequence and LwaCas13a G2 wt sensing crRNAs. Dark orange highlights the position of the 6 bp G2 wt sequence and crRNA spacers are shown in light orange. (**D**) Screen for G2 wt sensing LwaCas13a crRNAs detecting synthetic RNA containing target- (orange) and non-target- (blue) alleles. RNA detected at an overall concentration of 15 nM, each data point represents an individual reaction. RFUs are shown at 3 h divided by the mean RFU value of the non-template control. (**B–D**) Error bars: s.d. Statistical significance assessed with unpaired _t_ test; _P_ > 0.05= not significant (ns), *_P_ ≤ 0.05, **_P_ ≤ 0.01, ***_P_ ≤ 0.001 and ****_P_ ≤ 0.0001. For _P_ values see: Appendix Table S2. Source data are available online for this figure.

