## [Peer Review File · EMBO Molecular Medicine]

CRISPR-Enabled Point-of-Care Genotyping for APOL1 Genetic Risk Assessment

Robert Greensmith, Isadora Lape, Cristian Riella, Alexander Schubert, Jakob Metzger, Anand Dighe, Xiao Tan, Bernhard Hemmer, Josefine Rau, Sarah Wendlinger, Nora Diederich, Anja Schuetz, Leonardo Riella, and Michael Kaminski

Corresponding authors: Michael Kaminski (michael.kaminski@mdc-berlin.de) , Leonardo Riella (lriella@mgh.harvard.edu)

Review Timeline:

Submission Date:	22nd Feb 24
Editorial Decision:	22nd Mar 24
Revision Received:	14th Jun 24
Editorial Decision:	15th Jul 24
Revision Received:	19th Jul 24
Accepted:	12th Aug 24

Editor: Zeljko Durdevic

Transaction Report:

22nd Mar 2024

Dear Dr. Kaminski,

Thank you for the submission of your manuscript to EMBO Molecular Medicine. We have now received feedback from the two reviewers who agreed to evaluate your manuscript. As you will see from the reports pasted below, referee #1 is supportive, raises important but minor concerns and particularly highlights the methodology, clinical relevance, validation in more than 100 patients and comparison to the sequencing technology currently used for diagnostics. On the other hand, referee #2 recognizes interest of the study but also raises serious concerns particularly regarding the limited conceptual advance, low number of patients and the lack of information about the cost of reagents, personnel and setup costs when compared to sequencing technologies.

I have discussed these quite differing referee reports with my colleagues here and while we do agree with the referee #2 that the conceptual advance is somewhat limited, we also agree with referee #1 that the study has convincing clinical implications and we do appreciate validation of the assay in patient samples. Therefore, based on referee reports and given the translational implications of the study as well as it seems that all concerns are addressable, I would like to invite major revision of the current manuscript. Referee #2 concerns should be addressed by discussion, e.g. highlighting the conceptual advance of the study compared to previous reports and providing more detailed information in the manuscript text. Please add a table (or similar) with estimated costs comparison of your method and sequencing technology as suggested by the referee #2. If you would like to discuss further the points raised by the referees, I am available to do so via email or video. Let me know if you are interested in this option.

We would welcome the submission of a revised version within three months for further consideration. Please let us know if you require longer to complete the revision.

I look forward to receiving your revised manuscript.

Yours sincerely,

Zeljko Durdevic

We require:

- 1) A .docx formatted version of the manuscript text (including legends for main figures, EV figures and tables). Please make sure that the changes are highlighted to be clearly visible.
- 2) Individual production quality figure files as .eps, .tif, .jpg (one file per figure). For guidance, download the 'Figure Guide PDF': (<https://www.embopress.org/page/journal/17574684/authorguide#figureformat>).

3) A .docx formatted letter INCLUDING the reviewers' reports and your detailed point-by-point responses to their comments. As part of the EMBO Press transparent editorial process, the point-by-point response is part of the Review Process File (RPF), which will be published alongside your paper.

4) A complete author checklist, which you can download from our author guidelines (<https://www.embopress.org/page/journal/17574684/authorguide#submissionofrevisions>). Please insert information in the checklist that is also reflected in the manuscript. The completed author checklist will also be part of the RPF.

6) It is mandatory to include a 'Data Availability' section after the Materials and Methods. Before submitting your revision, primary datasets produced in this study need to be deposited in an appropriate public database, and the accession numbers and database listed under 'Data Availability'. Please remember to provide a reviewer password if the datasets are not yet public (see <https://www.embopress.org/page/journal/17574684/authorguide#dataavailability>).

13) Author contributions: You will be asked to provide CRediT (Contributor Role Taxonomy) terms in the submission system. These replace a narrative author contribution section in the manuscript.

14) A Conflict of Interest statement should be provided in the main text.

Please also suggest a striking image or visual abstract to illustrate your article as a PNG file 550 px wide x 300-800 px high.

***** Reviewer's comments *****

Referee #1 (Remarks for Author):

SUMMARY

Greensmith et al. report the development as well as extensive benchmarking and optimization of a novel diagnostic assay for multiplexed CRISPR-based genotyping of six clinically relevant APOL1 genotypes. They subsequently test and validate their assay in a clinical cohort across three clinical centers in the US, Germany, and Brazil, develop a machine-learning model for improved assay read-out, and finally engineer a lateral-flow based test that allows point-of-care application.

ASSESSMENT

This is a well-executed and highly relevant manuscript for multiple fields. First of all, it is centered on a new and exciting methodology that is hard to use and optimize. Furthermore, the authors have added an "edge" to their work by using CRISPR diagnostics (1) in a multiplexed fashion and (2), most importantly, they have found a clinically well-suited application of extremely high clinical relevance. This is why this work will also be of great interest to clinicians across multiple fields. Many previous CRISPR diagnostics papers were not convincing when it comes to the actual application, but this is a very good example of one or more potential use cases in the clinic. I especially liked the transplantation medicine angle, too. Finally, this is a breakthrough manuscript given the validation in >100 patient samples, essentially a small-scale "diagnostics clinical trial" on top. The rigorous characterization of weaknesses and concurrent optimization in a clinical setting, as well as the detailed comparison to state-of-the-art, i.e., sequencing, are very convincing.

This work will be transformative for the CRISPR diagnostics field and for clinical nephrology. Nonetheless, some minor questions and comments should be addressed before publication.

MINOR

1) Line 2 and across entire manuscript: Gene name APOL1 has to be in italics.

2) Across the entire manuscript: I do understand that some of the authors are of US origin, but I think we should stop using words like "race" and "Caucasian" in the scientific literature. These are not based on science and mostly reflect historic American nomenclature. I think ethnicity is OK and European ancestry etc.

3) Line 29-31: I don't get this sentence. Maybe you meant "while" instead of "although"?

4) Line 43: Cas enzymes generally don't have an "inherent single-nucleotide specificity". That depends on the enzyme and the

mismatch position within the (proto)spacer.

5) Line 70: PCR and sanger are not costly per se. I would rather focus on the time-issue, i.e., slow turn-around time of the conventional techniques. If PCR/Sanger is costly, it's only because of "clinical grade/GMP" conditions, but this will be the same for CRISPR diagnostics/RPA, right?

Line 96: Please show what the AA substitutions are. G342 to what, e.g., G342X?

Convention is usually to say what becomes what.

So if G342 is mutated to a codon of alanine, that would be G342A, for example.

6) Line 102: I wouldn't show mutation of A to G as "A:G" as the colon is usually used to show Watson/Crick/Frankling base-pairing, e.g., C:G and A:T.

Instead, I would write A>G

7) Line 104: I would mention that this is an in-frame deletion, specifically.

8) Line 110 and throughout manuscript/figures: I am not sure if the term "off-target" is the right choice to describe detection/sensing of the "unwanted" allele. Sorry, if I am misunderstanding this. But I think you use the term like this: If you want to discriminate let's say (1) WT allele from (2) the disease allele = allele of interest, then detection of the WT allele would be designated "off-target" detection, right?

If so, I think it is misleading because off-target to me sounds like binding/detection of a completely unrelated allele/locus (analogous to CRISPR gene editing). Whereas in your case both alleles are "on-target" with reference to the crRNA.

Maybe one can call them "target-allele" and "non-target allele"?

9) Line 112: you mention "position 3 of the spacer". Could you please briefly describe how you count, i.e, something like "with 1 being the most PFS/PAM-proximal/distal base".

10) Line 130: Please define RPA here, thanks.

11) Line 149: Could you please clarify here how you detect DNA targets with RNA-targeted Cas proteins like Cas13?

12) Line 239: The ML model is a bit "out of the blue". How does this work? Why is it simple? What's the input? Please provide the code in the supplement, too.

13) Lines 269-276: Was this done with patient samples, too? Sorry, if I missed this.

Would be great to show the lateral-flow assay also with a couple of patient samples in any case.

14) Line 492: Please add GraphPad version number.

15) Fig. 1a: Panel all the way to the right for normal/high risk genotypes shows 2x low-risk genotypes. For the red box you meant G1/G1, G1/G2, G2/G2?

16) Fig. 1c/e: see above re "on/off-target" terminology .

17) Fig. 3e: the blue stroke to mark the 6bp deletion was a bit hard to understand. Maybe put a box around the "future deletion" sequence or somehow mark it differently?

18) Fig. 3f: Here, on/off-target is reversed compared to the same variants in Fig. 1e.

19) Fig. 5a: In comparison to Fig. 4b, gDNA extraction is lacking here. Why? Still part of the workflow, isn't it? Would add some additional time, I'd think.

20) Fig. 6a: It would be very interesting to show the timeline using this assay. Probably very short, right? It's worth highlighting, I think.

Referee #2 (Remarks for Author):

Summary:

In this article the authors have used three different CRISPR-Cas enzymes LwaCas13a, PsmCas13b and LbaCas12a to develop

a point-of-care diagnostic platform to detect APOL1 disease variants. They have taken two disease-causing variants of APOL1 gene, G1 (A>G SNV) and G2 (delTTAT). For readout, they have used target-dependent collateral cleavage of fluorescently tagged RNA reporter molecules for LwaCas13a and PsmCas13b and direct dsDNA target cleavage for LbaCas12a. For RNA-based detection they first have used RPA to isothermally amplify the APOL1 target site and convert that amplicon to RNA. They have used a cohort of samples from 124 patients. Afterwards they have used genomic DNA from participants and a machine-learning model to validate the assay. They have shown the possibility of using the platform as a P.O.C device for APOL1 variant detection.

Considering that multiple studies in the past have used CRISPR Cas enzymes for developing POC platforms in combination with isothermal application and combinations of CRISPR systems have been utilized as well, the present study doesn't offer significant intellectual improvement over what is known in the field (Kumar et al. 2022, Mustafa et al. 2021, Kaminski et al. 2021 etc.). The number of patients reported (127) is also not very high to qualify this methodology as a very significant improvement over existing ones describing similar assays (Patschung et al. 2022). Importantly, the authors have mentioned that this method can provide POC, quick and cost effective solutions but no data has been shown comparing the cost of reagents, personnel and setup costs when placed side-by-side with Sequencing technologies. Thus, although the work going into standardization of assays for multiplexing different Cas proteins for genotyping is definitely appreciated, in the opinion of this reviewer, the improvements over existing technologies are not significant to warrant a publication in EMBO Mol. Med.

Minor Comments:

Line 82: "..robust analysis under various conditions." which conditions are not clear. Was it meant to indicate genotypes?

Line 84-85: "..While we demonstrate the use of CRISPR-genotyping for APOL1-mediated kidney disease", Did all the participants have kidney disease? What was the disease severity? Were they genotyped with sequencing?.

Line 130: Full abbreviation of "RPA assay" Recombinase Polymerase Amplification is not mentioned before.

Line 151-158: "We first tested the ability of the assay to detect and discriminate between synthetic DNA..", here it is not clear how they are measuring fluorescence with PsmCas13b with dsDNA, it needs mentioning of converting amplicon to RNA by T7 pol in the results section which is briefly mentioned in the methods section.

Line 269: The source of the DNA used for lateral flow assay is not clearly mentioned, it is unclear whether synthetic DNA or gDNA was used.

Fig3 line 594, "motives" should be changed to "motifs".

The authors have not used Sanger/

RESPONSES TO THE EDITOR

"We have now received feedback from the two reviewers who agreed to evaluate your manuscript. As you will see from the reports pasted below, referee #1 is supportive, raises important but minor concerns and particularly highlights the methodology, clinical relevance, validation in more than 100 patients and comparison to the sequencing technology currently used for diagnostics. On the other hand, referee #2 recognizes interest of the study but also raises serious concerns particularly regarding the limited conceptual advance, low number of patients and the lack of information about the cost of reagents, personnel and setup costs when compared to sequencing technologies.

I have discussed these quite differing referee reports with my colleagues here and while we do agree with the referee #2 that the conceptual advance is somewhat limited, we also agree with referee #1 that the study has convincing clinical implications and we do appreciate validation of the assay in patient samples. Therefore, based on referee reports and given the translational implications of the study as well as it seems that all concerns are addressable, I would like to invite major revision of the current manuscript. Referee #2 concerns should be addressed by discussion, e.g. highlighting the conceptual advance of the study compared to previous reports and providing more detailed information in the manuscript text. Please add a table (or similar) with estimated costs comparison of your method and sequencing technology as suggested by the referee #2".

We would like to thank the editor and reviewers for their time and effort spent on reviewing our manuscript. The feedback certainly has helped to improve our work. We have addressed all of the editor's and reviewers' comments. Specifically, we have better described the advance of our study to previous work in CRISPR-based genotyping (main text lines 306-318) and have added a table (EV. Table 3) comparing the costs, time and technical requirements between CRISPR-based APOL1 genotyping, Clinical Diagnostics APOL1 genotyping services, and sequencing. We have provided more detail on the specifics of the assay and the machine learning model. In addition, we have performed additional experiments validating the lateral flow assay on patient samples.

RESPONSES TO REVIEWER #1

1) Line 2 and across entire manuscript: Gene name APOL1 has to be in italics.

We changed all gene symbols to Italics.

2) Across the entire manuscript: I do understand that some of the authors are of US origin, but I think we should stop using words like "race" and "Caucasian" in the scientific literature. These are not based on science and mostly reflect historic American nomenclature. I think ethnicity is OK and European ancestry etc.

We changed "race" to "ethnicity" and replaced "Caucasian" with "European ancestry".

3) Line 29-31: I don't get this sentence. Maybe you meant "while" instead of "although"?

We replaced 'although' with 'while'.

4) Line 43: Cas enzymes generally don't have an "inherent single-nucleotide specificity". That depends on the enzyme and the mismatch position within the (proto)spacer.

We added extra information to this sentence, specifying that it is Cas12 or Cas13 enzymes, (Lines 43-44).

5) Line 70: PCR and sanger are not costly per se. I would rather focus on the time-issue, i.e., slow turn-around time of the conventional techniques. If PCR/Sanger is costly, it's only because of "clinical grade/GMP" conditions, but this will be the same for CRISPR diagnostics/RPA, right?

We removed the part of the sentence that describes these methods as costly, and referred to table EV3 for a detailed comparison of time, setup costs and point-of-care compatibility (Lines 68-70).

Line 96: Please show what the AA substitutions are. G342 to what, e.g., G342X? Convention is usually to say what becomes what. So if G342 is mutated to a codon of alanine, that would be G342A, for example.

We changed G342 to S342G throughout, (Lines 96, 98, 99).

6) Line 102: I wouldn't show mutation of A to G as "A:G" as the colon is usually used to show Watson/Crick/Frankling base-pairing, e.g., C:G and A:T. Instead, I would write A>G

We changed 'A:G' to 'A>G', (Line 102 and figures).

7) Line 104: I would mention that this is an in-frame deletion, specifically.

We included the specification that this is an in-frame deletion, (Line 104).

8) Line 110 and throughout manuscript/figures: I am not sure if the term "off-target" is the right choice to describe detection/sensing of the "unwanted" allele. Sorry, if I am misunderstanding this. But I think you use the term like this: If you want to discriminate let's say (1) WT allele from (2) the disease allele = allele of interest, then detection of the WT allele would be designated "off-target" detection, right? If so, I think it is misleading because off-target to me sounds like binding/detection of a completely unrelated allele/locus (analogous to CRISPR gene editing). Whereas in your case both alleles are "on-target" with reference to the crRNA. Maybe one can all them "target-allele" and "non-target allele"?

We changed the terms 'on-target' and 'off-target' to 'target-allele' and 'non-target-allele', respectively, throughout the text and in the relevant figures.

9) Line 112: you mention "position 3 of the spacer". Could you please briefly describe how you count, i.e, something like "with 1 being the most PFS/PAM-proximal/distal base".

We described how the positions are named, referencing position 1 as being most proximal to the crRNA direct repeat sequence, (Lines 112-113).

10) Line 130: Please define RPA here, thanks.

We defined RPA here as recombinase polymerase amplification, (Line 131).

11) Line 149: Could you please clarify here how you detect DNA targets with RNA-targeted Cas proteins like Cas13?

We added relevant information here to explain how the DNA amplicons are transcribed to RNA such that Cas13 can then detect them, (Lines 153-155).

12) Line 239: The ML model is a bit "out of the blue". How does this work? Why is it simple? What's the input? Please provide the code in the supplement, too.

We included more information that explains the motivation behind including the model for analysis, namely to eliminate a need for adjusting thresholds between experiments for genotype determination. We mention that the predictions are based on the signal intensities of the three fluorescent channels (inputs), (Lines 239-242). We have uploaded the code and the relevant data in the folder Source data expanded view folder Sourcedata_Expandedview.

13) Lines 269-276: Was this done with patient samples, too? Sorry, if I missed this. Would be great to show the lateral-flow assay also with a couple of patient samples in any case.

This was done on synthetic DNA and following your comment we have now also tested patient samples with the lateral flow readout. We completed this on three different patient samples for each of the six APOL1 genotypes. We included the new experimental results in an updated figure 6.

14) Line 492: Please add GraphPad version number.

We added the graphpad version number (8.4.3), (Line 500).

15) Fig. 1a: Panel all the way to the right for normal/high risk genotypes shows 2x low-risk genotypes. For the red box you meant G1/G1, G1/G2, G2/G2?

We have corrected this, such that the red box shows G1/G1, G1/G2, G2G2.

16) Fig. 1c/e: see above re "on/off-target" terminology.

We have adjusted the figure legend as per point 8 above.

17) Fig. 3e: the blue stroke to mark the 6bp deletion was a bit hard to understand. Maybe put a box around the "future deletion" sequence or somehow mark it differently?

We have put a red box around the deletion sequence, referenced in the figure legend.

18) Fig. 3f: Here, on/off-target is reversed compared to the same variants in Fig. 1e.

We have added more information to the text to explain why this is the case (lines 172-178). In summary, PsmCas13a crRNAs were screened to detect the wildtype allele for the G2 variant with the final aim of being able to discriminate between wt, heterozygous, and homozygous genotypes (Fig. 1e). Here, crRNA22 was the best performing, however when genotyping patient samples, it did not discriminate between wildtype (on-target) and heterozygous samples (Fig. 3d). Therefore, we introduced LbaCas12a to detect the opposite allele (mutant) for the G2 variant. With this, if the sample is wildtype then only PsmCas13b results in a high signal, while if the sample is mutant then only LbaCas12a results in a high signal. If the sample is heterozygous then both Cas enzymes result in a high signal. Thereby, analysing both PsmCas13b and LbaCas12's simultaneous cleavage activity enables discrimination of all three genotypes for the G2 variant.

19) Fig. 5a: In comparison to Fig. 4b, gDNA extraction is lacking here. Why? Still part of the workflow, isn't it? Would add some additional time, I'd think.

Yes, it is still part of the workflow. We have updated the figure to include genomic DNA extraction. As this information is relevant for Figs. 4d-e as well but represents redundant information it is now shown in a revised Fig. 4a and removed from Fig. 5. The investigation of collateral cleavage activity (previous Fig. 4a) was moved instead to Fig EV1.

20) Fig. 6a: It would be very interesting to show the timeline using this assay. Probably very short, right? It's worth highlighting, I think.

Visualization of the cleavage products on the lateral flow assay is 10 mins. The total time for RPA and CRISPR reactions is 2 hrs, similar to the fluorescence-based assay. We have clarified this timeline in the methods section (Lines 471-473).

RESPONSES TO REVIEWER #2

In this article the authors have used three different CRISPR-Cas enzymes LwaCas13a, PsmCas13b and LbaCas12a to develop a point-of-care diagnostic platform to detect APOL1 disease variants. They have taken two disease-causing variants of APOL1 gene, G1 (A>G SNV) and G2 (deITAT). For readout, they have used target-dependent collateral cleavage of fluorescently tagged RNA reporter molecules for LwaCas13a and PsmCas13b and direct dsDNA target cleavage for LbaCas12a. For RNA-based detection they first have used RPA to isothermally amplify the APOL1 target site and convert that amplicon to RNA. They have used a cohort of samples from 124 patients. Afterwards they have used genomic DNA from participants and a machine-learning model to validate the assay. They have shown the possibility of using the platform as a P.O.C device for APOL1 variant detection. Considering that multiple studies in the past have used CRISPR Cas enzymes for developing POC platforms in combination with isothermal application and combinations of CRISPR systems have been utilized as well, the present study doesn't offer significant intellectual improvement over what is known in the field (Kumar et al. 2022, Mustafa et al. 2021, Kaminski et al. 2021 etc.). The number of patients reported (127) is also not very high to qualify this methodology as a very significant improvement over existing ones describing similar assays (Patschung et al. 2022).

We clarified the novelty of our study as compared to previously published literature with a special focus on the studies pointed out by Reviewer #2 (Kumar et al. 2022, Mustafa et al. 2021, Kaminski et al. 2021 etc.). We have included additional information in the main text on lines 306-318 discussing the novelty of our assay.

Our study includes several significant advancements:

- Fully genotyping two variants (six genotypes) within a single multiplexed assay containing three Cas enzymes.
- Validating CRISPR-based genotyping in a relatively large clinical cohort of 124 patients.
- Demonstrating genotyping using a multi-analyte lateral flow readout.

As summarised in the three reviews, most current CRISPR-based diagnostic studies discriminating targets that differ by a single nucleotide have been focused on the detection of pathogens. These assays do not require the differentiation between wildtype, heterozygous, and homozygous necessary for human genotyping. Compared to pathogen detection, CRISPR-based genotyping of human samples is lacking validation in larger clinical cohorts, and we believe that our results thereby fill an important gap in the field.

In the following section, we discuss in more detail the 3 reviews highlighted by Reviewer #2: Unfortunately, we could not find the study Patschung et al. 2022 pointed out by Reviewer #2.

The focus of the review Kumar et al. 2021 is on the detection of single nucleotide variants (SNV) and Table 1 summarizes CRISPR-based assays developed to date. When considering the papers listed here, seven studies achieved human genotyping (Zhou et al. 2018, Balderston et al. 2021, Azhar et al. 2021, Teng et al. 2019, Harrington et al. 2018, Gootenberg et al. 2017, Gootenberg et al. 2018). Six of the studies used <10 clinical samples, while Azhar et al. 2021 validated their assay by genotyping 49 clinical samples for one SNV (the highest number of samples for human genotyping that we are aware of). Two of these studies achieved a lateral flow-based readout (Azhar et al. 2021 detecting SARS-CoV-2, Gootenberg et al. 2018 using <5 clinical samples), however not multi-analyte as in our study. Gootenberg et al. 2018 further demonstrated a multiplexed assay containing four Cas enzymes, however this assay only used synthetic standards mimicking viral detection. It was neither demonstrated on clinical samples nor for human genotyping.

Kaminski et al. 2021 refers to three additional studies that achieved human genotyping (Hajian et al. 2019, Li et al. 2018, Li L et al. 2019). Li et al. 2018 was the only study to include wildtype and heterozygous detection of a SNP, missing homozygous samples. Hajian et al. 2019 detected large scale fragment deletions in the X-chromosomal DMD gene of males, while it was not validated for genotyping of SNVs in autosomes. The studies also included lower numbers of clinical samples for validation than our study (Hajian et al. 2019 <10; Li L et al. 2019 <10; Li et al. 2018 <20), and did not include a lateral flow readout. Their follow-up paper detecting SNVs (Hajian et al. 2021) used one patient per genotype.

Mustafa et al. 2022 focuses on infectious disease detection. SNP detection is discussed, mainly relating to viruses. The two human genotyping studies mentioned here (Gootenberg et al. 2017, Gootenberg et al. 2018) have been discussed above.

Importantly, the authors have mentioned that this method can provide POC, quick and cost-effective solutions but no data has been shown comparing the cost of reagents, personnel and setup costs when placed side-by-side with Sequencing technologies. Thus, although the work going into standardization of assays for multiplexing different Cas proteins for genotyping is definitely appreciated, in the opinion of this reviewer, the improvements over existing technologies are not significant to warrant a publication in EMBO Mol. Med.

We have added a table (EV. Table 3) comparing the costs, time and technical requirements between CRISPR-based APOL1 genotyping and sequencing. We included a section *Hands-on time* rather than personnel costs, as these tend to be variable.

Minor Comments:

Line 82: "...robust analysis under various conditions." which conditions are not clear. Was it meant to indicate genotypes?

We added extra information to describe the conditions. Specifically, different test centres/ laboratories (with different personnel and machines), and also different reagent batches through the project, (Lines 80-82).

Line 84-85: "...While we demonstrate the use of CRISPR-genotyping for APOL1-mediated kidney disease", Did all the participants have kidney disease? What was the disease severity? Were they genotyped with sequencing?

We do not have information on the kidney disease status of all patients. However, all patients were genotyped with sequencing or qPCR. To better clarify we changed 'CRISPR-genotyping for APOL1-mediated kidney disease' to 'CRISPR-diagnostics for APOL1 genotyping' in the text (Line 84).

Line 130: Full abbreviation of "RPA assay" Recombinase Polymerase Amplification is not mentioned before.

We included 'recombinase polymerase amplification', (Line 131).

Line 151-158: "We first tested the ability of the assay to detect and discriminate between synthetic DNA..", here it is not clear how they are measuring fluorescence with PsmCas13b with dsDNA, it needs mentioning of converting amplicon to RNA by T7 pol in the results section which is briefly mentioned in the methods section.

We added relevant information here to explain how the DNA amplicons are transcribed to RNA such that Cas13 can then detect them, (Lines 153-155).

Line 269: The source of the DNA used for lateral flow assay is not clearly mentioned, it is unclear whether synthetic DNA or gDNA was used.

This was done on synthetic DNA. For the revised manuscript, we have also tested patient samples with the lateral flow readout. We completed this on three different patient samples for each of the six APOL1 genotypes and added the new experimental data (Fig. 6b).

Fig3 line 594, "motives" should be changed to "motifs".

We changed 'motives' to 'motifs' (Line 616).

The authors have not used Sanger/

Samples of the Brazil cohort (n=22 samples) were genotyped by qPCR as previously described (Riella C., et al., 2019), while samples of the USA and Germany cohorts (n=102 samples) were genotyped using PCR followed by Sanger sequencing (described in methods section *Sample preparation*; starting on line 390). We have now added this information also to lines 226-227 of the main text.

15th Jul 2024

Dear Dr. Kaminski,

Thank you for the submission of your revised manuscript to EMBO Molecular Medicine. I am pleased to inform you that we will be able to accept your manuscript pending the following final amendments:

1) In the main manuscript file, please do the following:

- Please address all comments suggested by our data editors listed below:

o Figure legends:

1. Please note that the exact p values are not provided in the legends of figures 1c, e; 2a-c; 3f; EV 4b, d.

2. Please note that in figures 1c, e; 2a-c; 3f; there is a mismatch between the annotated p values in the figure legend and the annotated p values in the figure file that should be corrected.

3. Please note that the box plots need to be defined in terms of minima, maxima, centre, bounds of box and whiskers, and percentile in the legends of figures 5a; EV 3a.

4. Please note that information related to n is missing in the legends of figures 4d-e.

5. Please note that the error bars are not defined in the legends of figures 2a-d; 4d-e.

- The manuscript sections should be in the following order: Title page - Abstract & Keywords - Introduction - Results - Discussion - Methods - Data Availability - Acknowledgments - Disclosure Statement & Competing Interests - References - Figure Legends - (Main Tables with legends) - Expanded View Figure Legends.

- Add up to 5 keywords.

- Rename "Competing interests" to "Disclosure and competing interests statement". We updated our journal's competing interests policy in January 2022 and request authors to consider both actual and perceived competing interests. Please review the policy <https://www.embopress.org/competing-interests> and update your competing interests if necessary.

- Author contributions: Please remove it from the manuscript and specify author contributions in our submission system. CRediT has replaced the traditional author contributions section because it offers a systematic machine-readable author contributions format that allows for more effective research assessment. You are encouraged to use the free text boxes beneath each contributing author's name to add specific details on the author's contribution. More information is available in our guide to authors:

<https://www.embopress.org/page/journal/17574684/authorguide#authorshipguidelines>

- Please include structured Methods section that includes a Reagents and Tools Table followed by a Methods and Protocols section. More information on how to adhere to this format as well as downloadable templates (.docx) for the Reagents and Tools Table can be found in our author guidelines: <https://www.embopress.org/page/journal/17574684/authorguide#structuredmethods> An example of a paper with Structured Methods can be found here:

<https://www.embopress.org/doi/full/10.1038/s44320-024-00037-6#sec-4>

- In Methods, provide the statement that informed consent was obtained from all human subjects and confirm that the experiments conformed to the principles set out in the WMA Declaration of Helsinki and the Department of Health and Human Services Belmont Report.

- In Methods, a statistical paragraph should reflect all information that you have filled in the Authors Checklist, especially regarding randomization, blinding, replication.

- Indicate in legends number and nature of replicates and exact p= values, not a range, along with the statistical test used. To keep the figures "clear" some authors found providing an Appendix table Sx with all exact p-values preferable. You are welcome to do this if you want to.

- Correct the reference citation in the text and reference list. In the text a reference should be cited by author and year of publication. Include a space between a word and the opening parenthesis of the reference that follows. In the reference list, citations should be listed in alphabetical order. Where there are more than 10 authors on a paper, 10 will be listed, followed by "et al.". Also, please remove DOIs. Please check "Author Guidelines" for more information.

<https://www.embopress.org/page/journal/17574684/authorguide#referencesformat>

- In data availability statement please remove the first sentence and only leave "This study includes no data deposited in external repositories."

2) Appendix: Please rename "Expanded view" file to "Appendix", remove all EV figures and rename EV Tables to Appendix Table S1, etc.. Add table of content on the title page and upload it as a PDF file. Please add updated callouts for all Appendix Tables in the main manuscript text.

3) Funding: Please make sure that information about all sources of funding are complete in both our submission system and in the manuscript. BIH Charité Clinician Scientist Program funded by the Charité - Universitätsmedizin Berlin and the Berlin Institute of Health at Charité (BIH), BIH MD Research Stipend are currently missing in our submission system.

4) The Paper Explained: Please rename "Summary" to "The Paper Explained" and add it to the main manuscript file with following subheadings "Problem", "Results" and "Impact".

5) Synopsis:

- Synopsis image: Please resize the image to 550 px-wide x (250-400)-px high and upload it as a high-resolution jpeg file. Also,

please increase font size in the image.

6) For more information: This space should be used to list relevant web links for further consultation by our readers. Could you identify some relevant ones and provide such information as well? Some examples are patient associations, relevant databases, OMIM/proteins/genes links, author's websites, etc...

7) As part of the EMBO Publications transparent editorial process initiative (see our Editorial at <http://embomolmed.embopress.org/content/2/9/329>), EMBO Molecular Medicine will publish online a Review Process File (RPF) to accompany accepted manuscripts. This file will be published in conjunction with your paper and will include the anonymous referee reports, your point-by-point response and all pertinent correspondence relating to the manuscript. Let us know whether you agree with the publication of the RPF and as here, if you want to remove or not any figures from it prior to publication. Please note that the Authors checklist will be published at the end of the RPF.

8) Please provide a point-by-point letter INCLUDING my comments as well as the reviewer's reports and your detailed responses (as Word file).

I look forward to reading a new revised version of your manuscript as soon as possible.

Yours sincerely,

Zeljko Durdevic

*** Instructions to submit your revised manuscript ***

1) a .docx formatted version of the manuscript text (including Figure legends and tables)

2) Separate figure files*

3) supplemental information as Expanded View and/or Appendix. Please carefully check the authors guidelines for formatting Expanded view and Appendix figures and tables at <https://www.embopress.org/page/journal/17574684/authorguide#expandedview>

4) a letter INCLUDING the reviewer's reports and your detailed responses to their comments (as Word file).

5) The paper explained: EMBO Molecular Medicine articles are accompanied by a summary of the articles to emphasize the major findings in the paper and their medical implications for the non-specialist reader. Please provide a draft summary of your article highlighting

6) For more information: There is space at the end of each article to list relevant web links for further consultation by our readers. Could you identify some relevant ones and provide such information as well? Some examples are patient associations, relevant databases, OMIM/proteins/genes links, author's websites, etc...

7) Author contributions: the contribution of every author must be detailed in a separate section.

8) EMBO Molecular Medicine now requires a complete author checklist (<https://www.embopress.org/page/journal/17574684/authorguide>) to be submitted with all revised manuscripts. Please use the checklist as guideline for the sort of information we need WITHIN the manuscript. The checklist should only be filled with page numbers where the information can be found. This is particularly important for animal reporting, antibody dilutions (missing) and exact values and n that should be indicated instead of a range.

9) Every published paper now includes a 'Synopsis' to further enhance discoverability. Synopses are displayed on the journal webpage and are freely accessible to all readers. They include a short stand first (maximum of 300 characters, including space) as well as 2-5 one sentence bullet points that summarise the paper. Please write the bullet points to summarise the key NEW findings. They should be designed to be complementary to the abstract - i.e. not repeat the same text. We encourage inclusion of key acronyms and quantitative information (maximum of 30 words / bullet point). Please use the passive voice. Please attach these in a separate file or send them by email, we will incorporate them accordingly.

You are also welcome to suggest a striking image or visual abstract to illustrate your article. If you do please provide a jpeg file 550 px-wide x 300-600px high.

10) A Conflict of Interest statement should be provided in the main text

11) Please note that we now mandate that all corresponding authors list an ORCID digital identifier. This takes <90 seconds to complete. We encourage all authors to supply an ORCID identifier, which will be linked to their name for unambiguous name identification.

Currently, our records indicate that the ORCID for your account is 0000-0003-0429-7027.

Link Not Available

12) Include a Reagents and Tools Table as part of the Methods section, which can be downloaded from our author guidelines (<https://www.embopress.org/page/journal/17574684/authorguide#structuredmethods>)

Photos 400-800 DPI

*Additional important information regarding figures and illustrations can be found at

<https://bit.ly/EMBOPressFigurePreparationGuideline>. See also figure legend preparation guidelines:

<https://www.embopress.org/page/journal/17574684/authorguide#figureformat>

***** Reviewer's comments *****

Referee #1 (Remarks for Author):

Thanks for addressing all my comments and concerns. The manuscript has been improved considerably, especially with regard to EV table 3 comparing cost/time and further validation of the lateral flow assay on patient samples. This is high quality work that is important for the field and of high interest to the broad readership of EMM.

The authors addressed the remaining editorial issues.

12th Aug 2024

Dear Dr. Kaminski,

We are pleased to inform you that your manuscript is accepted for publication and is now being sent to our publisher to be included in the next available issue of EMBO Molecular Medicine.
